JCB — Journal of Cell Biology

## REPORT

# NCOA4 drives ferritin phase separation to facilitate macroferritinophagy and microferritinophagy

Tomoko Ohshima[1]*, Hayashi Yamamoto[1,2]*, Yuriko Sakamaki[3], Chieko Saito[1], and Noboru Mizushima[1]

A ferritin particle consists of 24 ferritin proteins (FTH1 and FTL) and stores iron ions within it. During iron deficiency, ferritin particles are transported to lysosomes to release iron ions. Two transport pathways have been reported: macroautophagy and ESCRT-dependent endosomal microautophagy. Although the membrane dynamics of these pathways differ, both require NCOA4, which is thought to be an autophagy receptor for ferritin. However, it is unclear whether NCOA4 only acts as an autophagy receptor in ferritin degradation. Here, we found that ferritin particles form liquid-like condensates in a NCOA4-dependent manner. Homodimerization of NCOA4 and interaction between FTH1 and NCOA4 (i.e., multivalent interactions between ferritin particles and NCOA4) were required for the formation of ferritin condensates. Disruption of these interactions impaired ferritin degradation. Time-lapse imaging and three-dimensional correlative light and electron microscopy revealed that these ferritin–NCOA4 condensates were directly engulfed by autophagosomes and endosomes. In contrast, TAX1BP1 was not required for the formation of ferritin–NCOA4 condensates but was required for their incorporation into autophagosomes and endosomes. These results suggest that NCOA4 acts not only as a canonical autophagy receptor but also as a driver to form ferritin condensates to facilitate the degradation of these condensates by macroautophagy (i.e., macroferritinophagy) and endosomal microautophagy (i.e., microferritinophagy).

## Introduction

Iron is essential for various biological processes, but in excess it can cause deleterious effects (Bogdan et al., 2016; Fenton, 1894; Haber and Weiss, 1934; Halliwell and Gutteridge, 1990; Pantopoulos et al., 2012). Thus, the intracellular iron level must be tightly regulated. Ferritin is a key player in the regulation of iron homeostasis and is highly conserved among organisms except for some fungi (e.g., yeasts) and several bacterial and archaeal species (Bai et al., 2015; Canessa and Larrondo, 2013). It forms a particle consisting of 24 subunits of ferritin heavy chain 1 (FTH1) and ferritin light chain (FTL), which incorporates up to 4,500 $Fe^{2+}$ ions, oxidizes them to $Fe^{3+}$, and stores them as mineral cores in its hollow cavity (Mann et al., 1986). During iron deficiency, ferritin particles are transported to lysosomes in order to release the stored iron atoms (Asano et al., 2011; Kidane et al., 2006). Ferritin delivery to lysosomes is mediated by two different pathways: macroautophagy (Asano et al., 2011; Dowdle et al., 2014; Goodwin et al., 2017; Mancias et al., 2014) and the endosomal sorting complexes required for the transport (ESCRT)-mediated pathway (Goodwin et al., 2017). To degrade ferritin, both pathways require nuclear receptor coactivator 4 (NCOA4), currently considered to be a ferritin receptor, and the

macroautophagy adaptor Tax1-binding protein 1 (TAX1BP1; Dowdle et al., 2014; Goodwin et al., 2017; Mancias et al., 2014). Because NCOA4 and TAX1BP1 are degraded by not only macroautophagy but also endosomal microautophagy (Mejlvang et al., 2018), the ESCRT-dependent ferritin degradation reported by Goodwin et al. (2017) is likely mediated by microautophagy. Although NCOA4 has been shown to interact with TAX1BP1 (Goodwin et al., 2017), little is known about how it is involved in ferritin degradation through the morphologically and mechanistically distinct pathways of macroautophagy and endosomal microautophagy.

Electron microscopy studies in cultured cells and tissues have demonstrated that ferritin particles can be clustered in the cytosol (Heynen and Verwilghen, 1982; Takano-Ohmuro et al., 2000) as well as within membranous structures (Heynen and Verwilghen, 1982; Iancu et al., 2014; Sullivan et al., 1976). Furthermore, our group previously found that clusters of ferritin particles are enlarged in macroautophagy-deficient cells, some of which are >500 nm in size, and accumulate at autophagosome formation sites (Kishi-Itakura et al., 2014). These ferritin clusters are spherical and often associate with SQSTM1 (also called

[1]Department of Biochemistry and Molecular Biology, Graduate School and Faculty of Medicine, The University of Tokyo, Tokyo, Japan; [2]Department of Molecular Oncology, Institute for Advanced Medical Sciences, Nippon Medical School, Tokyo, Japan; [3]Microscopy Research Support Unit, Research Core, Tokyo Medical and Dental University, Tokyo, Japan.

*T. Ohshima and H. Yamamoto contributed equally to this paper. Correspondence to Noboru Mizushima: nmizu@m.u-tokyo.ac.jp.

p62) bodies without mixing their contents. Based on these findings, we hypothesized that ferritin clusters are liquid-like condensates caused by liquid–liquid phase separation (LLPS).

In this study, we show that ferritin clusters indeed have liquid-like properties and that NCOA4 is required for ferritin phase separation. The ferritin–NCOA4 condensates are incorporated into autophagosomes and endosomes in a TAX1BP1-dependent manner. Failure of condensate formation impairs ferritin degradation. These data suggest that the formation of liquid-like ferritin–NCOA4 condensates is a common mechanism to facilitate degradation by both macroautophagy and endosomal microautophagy.

## Results and discussion

### Ferritin particles assemble to form large condensates that exhibit liquid-like properties

To observe the distribution of ferritin particles in living cells, we established HeLa cells stably expressing monomeric EGFP (mGFP)-tagged FTH1 and FTL. Both mGFP-FTH1 and mGFP-FTL formed punctate structures in the cytoplasm in WT cells under normal (Fig. 1 A) and iron-replete conditions that induced ferritin expression (Fig. 1 B). In addition, mGFP-FTH1 colocalized with endogenous FTL puncta (Fig. S1 A). The expression levels of mGFP-FTH1 and mGFP-FTL were comparable to (or even lower than) those of endogenous FTH1 and FTL (Fig. 1 C), suggesting that the formation of puncta was not due to overexpression. The ferritin puncta became larger in macroautophagy-deficient FIP200 KO cells (Fig. 1, A and B). Transmission electron microscopy (TEM) of FIP200 KO cells expressing GFP-FTH1 showed cytosolic spherical clusters of the electron-dense ferritin particles (Fig. 1 D), consistent with our previous report (Kishi-Itakura et al., 2014). In addition, the clustered ferritin structures were not enclosed by membranes (Fig. 1 D, right panel). These structural characteristics (i.e., the highly spherical shape and the lack of membranes) raised the possibility that the ferritin clusters are biomolecular condensates driven by LLPS. Fluorescence recovery after photobleaching (FRAP) measurements in WT cells revealed that mGFP-FTH1 condensates exhibited ∼20% recovery within 10 min (Fig. 1, E and F; and Fig. S1 B) and mGFP-FTL condensates also exhibited fluorescence recovery after photobleaching (Fig. S1 C). In addition, coalescence of two discrete GFP-FTH1 condensates was observed (Fig. 1 G). These features, together with the abovementioned structural characteristics, are consistent with the current criteria for liquid-like biomolecular condensates formed via LLPS (Alberti et al., 2019). Thus, these results suggest that ferritin particles have the propensity to congregate to form liquid-like biomolecular condensates.

### NCOA4 drives ferritin phase separation

NCOA4 is involved in ferritin delivery to lysosomes by direct interaction with FTH1 (Dowdle et al., 2014; Mancias et al., 2014; Mancias et al., 2015). Thus, we investigated the relationship between ferritin condensates and NCOA4. In WT and FIP200 KO cells, mGFP-NCOA4 colocalized with the mRuby3-FTH1 puncta (Fig. 2 A) and HaloTag7-tagged NCOA4 (Halo-NCOA4) colocalized

with endogenous FTL puncta (Fig. S1 D), indicating that NCOA4 is a component of ferritin condensates. Then, to investigate the role of NCOA4 in condensate formation, we generated NCOA4 KO cells by using the CRISPR-Cas9 method (Fig. S1 E). The resulting cells showed a defect in the degradation of FTH1 upon iron depletion, achieved via the iron chelator deferoxamine (DFO; Fig. S1 F), which was in line with the previous reports (Dowdle et al., 2014; Mancias et al., 2014). Fluorescence microscopy revealed that mGFP-FTH1, which formed punctate structures in WT and FIP200 KO cells, became diffuse in the absence of NCOA4 (Fig. 2, B and C), suggesting that NCOA4 is indispensable for the formation of ferritin condensates.

Next, we examined whether FTH1 and FTL were necessary for the formation of ferritin–NCOA4 condensates. Knockdown of FTL did not impede mGFP-NCOA4 puncta formation in FIP200 KO cells (Fig. 2, D and E), suggesting that FTL is not required for the formation of NCOA4 condensate. In contrast, knockdown of FTH1 inhibited mGFP-NCOA4 puncta formation. However, as it also reduced the expression levels of mGFP-NCOA4, it was difficult to evaluate the specific role of FTH1 (Fig. 2, D and E).

Given that FTH1 interacts with NCOA4 (Mancias et al., 2015), we further investigated whether FTH1 and NCOA4 were sufficient for the formation of ferritin condensates. When muGFP-tagged human NCOA4 was exogenously expressed in yeast cells (ferritin and NCOA4 homologs are not present in yeast), it mostly dispersed in the cytoplasm and occasionally formed some small dots. In contrast, when coexpressed with FTH1, muGFP-NCOA4 formed one large punctum per cell (Fig. 2 F). These results suggest that FTH1 and NCOA4 are sufficient for the formation of ferritin–NCOA4 condensates.

### Ferritin–NCOA4 condensate formation is driven by NCOA4 self-interaction and NCOA4-FTH1 interaction

NCOA4 consists of the N-terminal coiled-coil domain (N), middle domain (M), and C-terminal domain (C), which are interconnected with intrinsically disordered regions (IDRs; IDR1 and IDR2; Fig. 3 A). To determine which domain(s) of NCOA4 is required for the formation of ferritin condensates, we constructed domain truncation mutants (Fig. 3 A). NCOA4 KO cells expressing 3×FLAG-tagged NCOA4 (FLAG-NCOA4) restored the formation of mGFP-FTH1 puncta (Fig. 3, B and C). The ΔM and ΔC mutants of FLAG-NCOA4 also recovered the formation of puncta. In contrast, the ΔN, ΔIDR1, and ΔIDR2 mutants failed to form mGFP-FTH1 puncta (Fig. 3, B and C). These results indicate the importance of the N, IDR1, and IDR2 domains of NCOA4 in the formation of ferritin condensates, which could be partly explained by the interaction of IDR2 with FTH1 (Gryzik et al., 2017; Mancias et al., 2015).

The N-terminal coiled-coil domain of NCOA4 is known as a self-oligomerization domain (Monaco et al., 2001). In fact, full-length FLAG-NCOA4 interacted with the N-terminal domain of NCOA4 (NCOA4N) fused with mGFP (Fig. 3 D). The HHpred search (Soding et al., 2005) identified that the N-terminal coiled-coil domain in NCOA4 is structurally similar to that in TRIM28. TRIM28 forms a homodimer (Stoll et al., 2019), and Ile-299 and Leu-306 are located at the homodimerization interface (Fig. S2 A). Thus, we introduced mutations in the

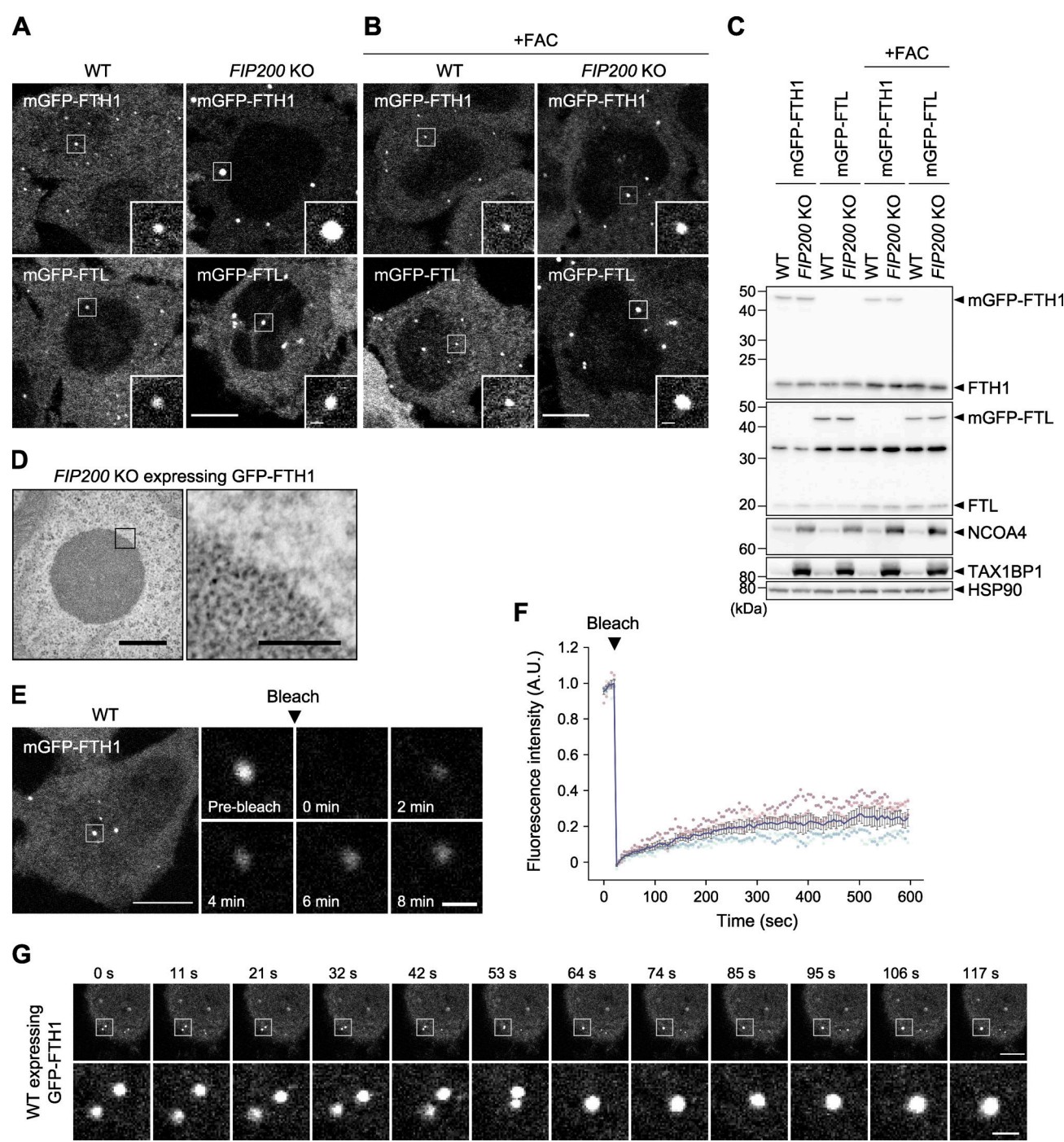

Figure 1. **Ferritin particles assemble to form liquid-like condensates. (A and B)** Fluorescent images of the ferritin subunits FTH1 and FTL. WT and macroautophagy-deficient *FIP200* KO HeLa cells stably expressing mGFP-FTH1 or mGFP-FTL were grown in DMEM (A) or treated with 10 μg/ml ferric ammonium citrate (FAC) for 24 h (B) and observed by fluorescence microscopy. Scale bars, 10 μm (main) and 1 μm (inset). **(C)** Immunoblots showing the expression levels of mGFP-FTH1 and mGFP-FTL in cells used in A. **(D)** TEM images of the ferritin condensates in *FIP200* KO HeLa cells expressing GFP-FTH1. Scale bars, 500 nm (main) and 100 nm (magnified). **(E)** FRAP analyses of the ferritin condensates. WT HeLa cells expressing mGFP-FTH1 were treated with 100 μg/ml FAC for 24 h followed by 50 μM deferoxamine (DFO) for 5 h, and subjected to FRAP analyses. Scale bars, 10 μm (main) and 1 μm (magnified). **(F)** Quantification of E. Fluorescence intensities before photobleaching were set to 1. Representative results from two independent experiments are presented as means ± SEM (*n* = 5). **(G)** Coalescence of the ferritin condensates. WT HeLa cells expressing GFP-FTH1 were grown in DMEM and observed by time-lapse fluorescence microscopy at ~11-s intervals. Scale bars, 10 μm (main) and 1.5 μm (magnified). Source data are available for this figure: SourceData F1.

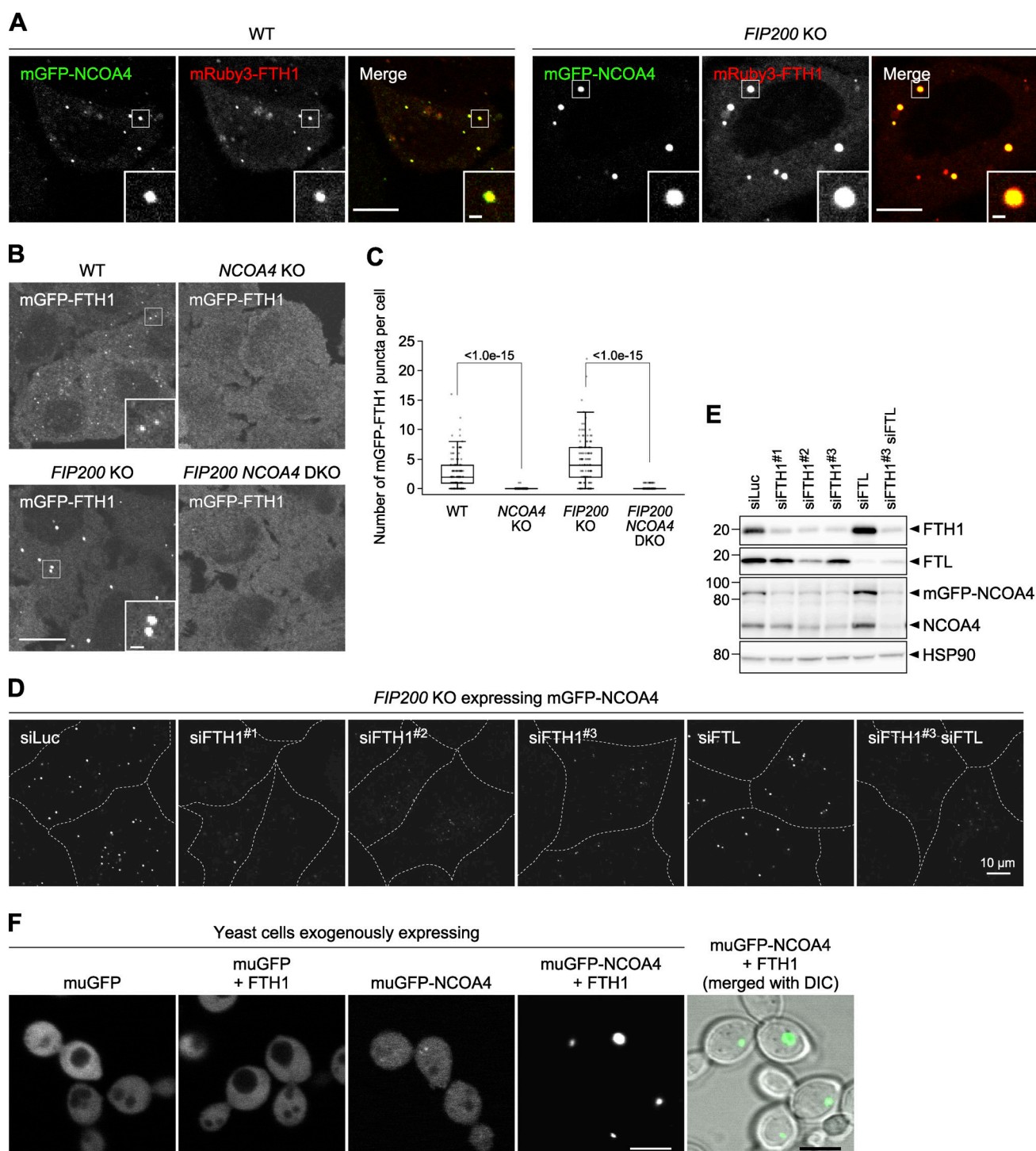

Figure 2. **NCOA4 drives ferritin condensate formation. (A)** Localization of NCOA4 in ferritin condensates. WT and *FIP200* KO HeLa cells expressing both mGFP-NCOA4 and mRuby3-FTH1 were grown in DMEM and observed by fluorescence microscopy. Scale bars, 10 μm (main) and 1 μm (inset). **(B)** NCOA4 is essential for condensate formation. WT, *NCOA4* KO, *FIP200* KO, and *FIP200 NCOA4* double KO (DKO) HeLa cells expressing mGFP-FTH1 were observed as in A. Scale bars, 15 μm (main) and 1.5 μm (inset). **(C)** The number of mGFP-FTH1 puncta per cell in B was quantified (*n* = 112–210 cells from three biological replicates). Solid bars indicate the means, boxes the interquartile ranges, and whiskers the 10th to 90th percentiles. Differences of puncta counts among the cells were statistically analyzed by Welch's *t* test with Holm–Sidak method for multiple comparison (two-tailed test). Data distribution was assumed to be normal, which was not formally tested. **(D)** Knockdown of ferritin subunits. *FIP200* KO HeLa cells expressing mGFP-NCOA4 were transfected with the indicated siRNAs (siLuc, siFTH1 [three independent oligos were used], and siFTL) and observed by fluorescence microscopy. Scale bar, 10 μm. **(E)** Immunoblots showing the expression levels of FTH1, FTL, and NCOA4 in cells used in D. **(F)** NCOA4 and FTH1 are sufficient for condensate formation. Yeast cells exogenously expressing muGFP-NCOA4 with or without FTH1 were observed by fluorescence microscopy. Scale bar, 5 μm. Source data are available for this figure: SourceData F2.

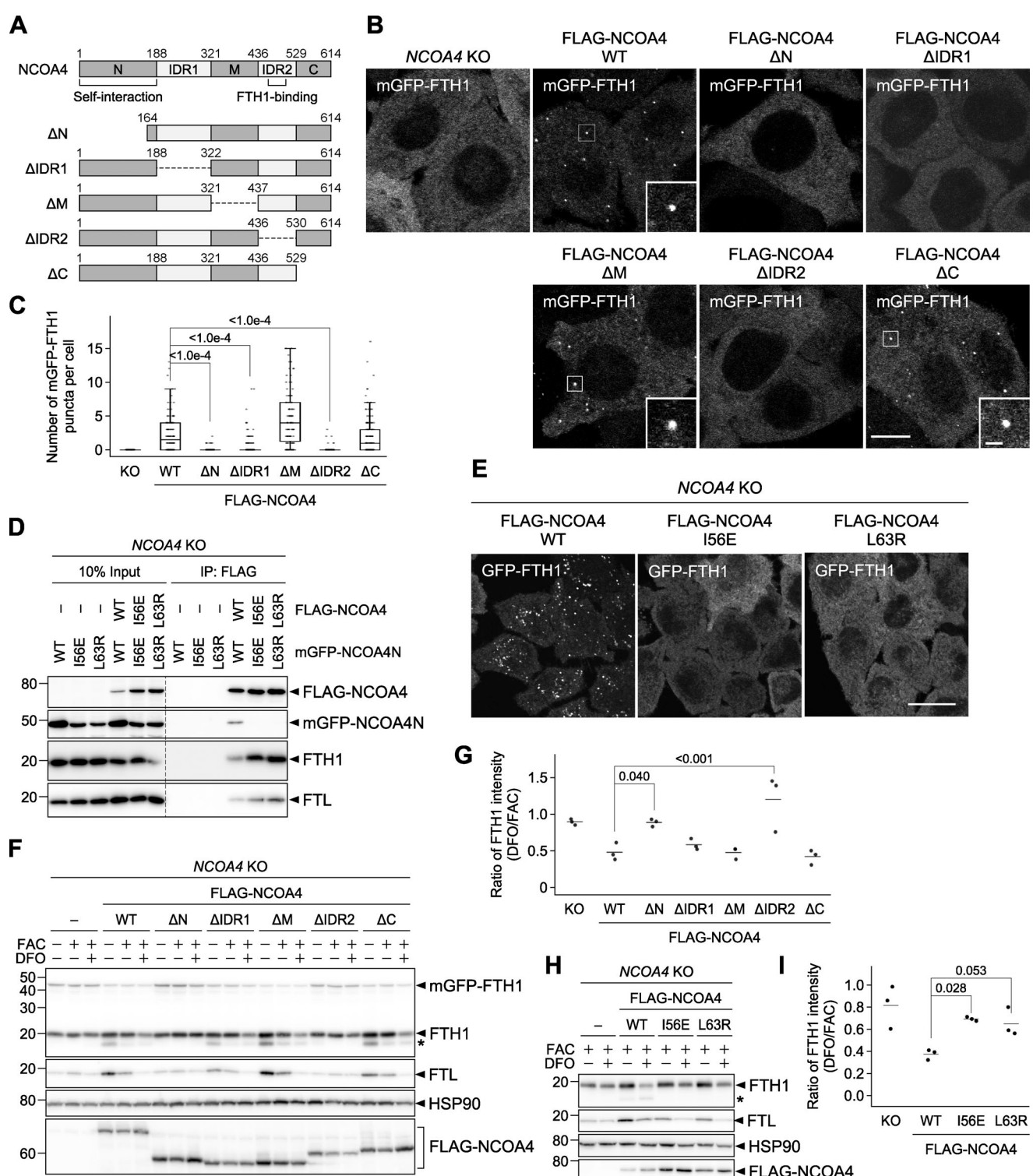

Figure 3. **Ferritin–NCOA4 condensate formation is driven by NCOA4 self-interaction and responsible for ferritin degradation. (A)** Truncated constructs of NCOA4. IDR, intrinsically disordered region. **(B)** WT or the truncated mutants of FLAG-NCOA4 were stably expressed in *NCOA4* KO HeLa cells harboring mGFP-FTH1. Scale bars, 10 μm (main) and 1.5 μm (inset). **(C)** Numbers of mGFP-FTH1 puncta per cell in B were counted (*n* = 110–151 cells from three biological replicates). Solid bars indicate the means, boxes the interquartile ranges, and whiskers the 10th to 90th percentiles. Differences among the cells expressing FLAG-NCOA4 were statistically analyzed by Dunnett's multiple comparison test (two-tailed test). Data distribution was assumed to be normal, which was not formally tested. **(D)** Defective self-interaction of the N-terminal domain mutants of NCOA4. *NCOA4* KO HeLa cells expressing both FLAG-NCOA4 (full-length) and mGFP-NCOA4N (the N-terminal domain of NCOA4) with or without I56E or L63R mutation were subjected to coimmunoprecipitation using anti-FLAG antibody. The samples were analyzed by immunoblotting with the antibodies against FLAG, GFP, FTH1, and FTL. **(E)** The I56E or L63R mutant of FLAG-NCOA4 was stably expressed in *NCOA4* KO HeLa cells harboring GFP-FTH1. The cells were observed as in A. **(F)** Cells used in B were grown in DMEM and treated with 10 μg/ml FAC for 24 h followed by 50 μM DFO for 12 h. Whole-cell lysates were analyzed by immunoblotting with the antibodies against FTH1,

FTL, HSP90, and FLAG. Asterisk represents a putative degradation product of FTH1. **(G)** FTH1 degradation upon DFO treatment in F. The ratio of the endogenous FTH1 band intensities under DFO-treated conditions to those under FAC-treated conditions is shown. Solid bars indicate the means, and dots the data from three independent experiments. Differences were statistically analyzed by Dunnett's multiple comparison test (two-tailed test). Data distribution was assumed to be normal, which was not formally tested. **(H)** The cells used in E were examined as in F. **(I)** Quantification of H as in G. Source data are available for this figure: SourceData F3.

corresponding residues in NCOA4 (I56E and L63R), which were predicted via AlphaFold-Multimer (Evans et al., 2022; Mirdita et al., 2022; Fig. S2 B). These mutations blocked self-interaction via the N-terminal domain (Fig. 3 D). The yeast two-hybrid assay using NCOA4N also showed that the I56E or L63R mutants were defective in self-interaction (Fig. S2 C). Fluorescence microscopy revealed that the I56E or L63R mutants did not show GFP-FTH1 puncta in *NCOA4* KO cells (Fig. 3 E), indicating that NCOA4 self-interaction is required for the formation of ferritin–NCOA4 condensates. Taken together, we concluded that the formation of ferritin–NCOA4 condensates is driven by NCOA4-mediated multivalent interactions (i.e., NCOA4-FTH1 interaction and NCOA4 self-interaction).

### Ferritin–NCOA4 condensate formation is required for ferritin degradation

We then determined whether condensate formation is required for the degradation of ferritin. When introduced into *NCOA4* KO cells, WT and all the NCOA4 mutants that restored the formation of mGFP-FTH1 puncta (ΔM, ΔC, and ΔIDR1) also rescued FTH1 degradation upon iron depletion, whereas the ΔN and ΔIDR2 mutants, which failed to form mGFP-FTH1 puncta, also failed to degrade FTH1 (Fig. 3, F and G). Furthermore, the dimerization-defective NCOA4 mutants (I56E and L63R) failed to restore FTH1 degradation (Fig. 3, H and I). It should be noted that these I56E and L63R mutants retained the binding ability to FTH1 and FTL (likely through FTH1 in ferritin particles; Fig. 3 D). Thus, these results suggest that NCOA4 dimerization-mediated condensate formation is important for ferritin degradation.

### Ferritin–NCOA4 condensates are common substrates for macroautophagy and endosomal microautophagy

Next, we investigated whether ferritin–NCOA4 condensates were targeted by autophagosomes. WT cells expressing mGFP-FTH1 and HaloTag7-tagged LC3B (Halo-LC3), an autophagosomal membrane marker (Kabeya et al., 2000; Mizushima, 2004), were observed under iron-deficient conditions. Time-lapse fluorescence microscopy showed that some mGFP-FTH1 puncta were sequestered by cup-shaped autophagosomal membranes in a piecemeal manner (Fig. 4 A and Video 1). To obtain more detailed morphological information, we conducted three-dimensional correlative light and electron microscopy (3D-CLEM). In line with the observations under a fluorescence microscope, we observed an electron-dense ferritin condensate with a diameter of ~500 nm being sequestered partly by a Halo-LC3-positive autophagosomal membrane (Fig. 4 B). The autophagosomal membrane appeared to be in close contact with the ferritin condensate, similar to the fluidophagy of the SQSTM1 bodies (Agudo-Canalejo et al., 2021). The remaining region not engulfed by the autophagosomal membrane retained a spherical

shape, consistent with its liquid-like property. These results suggest that ferritin–NCOA4 condensates are selective substrates for macroautophagy.

We also examined the involvement of endosomal microautophagy. To enlarge endosomes so that we could see intraluminal vesicles (ILVs) by fluorescence microscopy, we took advantage of mRuby3-RAB5$^{Q79L}$, a constitutively active mutant of RAB5 (Mejlvang et al., 2018; Stenmark et al., 1994). After doxycycline-induced expression of mRuby3-RAB5$^{Q79L}$, some GFP-FTH1 puncta were seen trapped in enlarged endosomes even under normal growth conditions (Fig. 4 C, upper panels). Similar results were obtained when mGFP-NCOA4 was used (Fig. 4 C, middle panels). The magnified images showed that the fluorescence intensity of the GFP-FTH1 puncta inside the enlarged endosomes (Fig. 4 C, lower panels, arrows) was similar to that in the cytosol (Fig. 4 C, lower panels, arrowhead), suggesting that ferritin condensates were directly incorporated into endosomes. These puncta in enlarged endosomes moved quickly (Videos 2, 3, and 4). Some diffuse GFP signals were detected inside endosomes, likely representing the disruption of ILV membranes (Fig. 4 C). In addition, 3D-CLEM using scanning electron microscopy (SEM) revealed that the enlarged endosomes containing mGFP-FTH1 puncta were electron dense and contained electron-dense ILVs (Fig. 4 D). By TEM, ferritin particles were detected in these ILVs (Fig. 4 E). In contrast to WT cells, *NCOA4* KO cells did not incorporate mGFP-FTH1 into enlarged endosomes at all (Fig. 4 F), suggesting that the formation of NCOA4-mediated condensates is required for the incorporation of ferritin into endosomes. Consistent with this, the ΔN, ΔIDR2, I56E, and L63R mutants of NCOA4, which failed to form mGFP-FTH1 puncta (Fig. 3, B and C), failed to incorporate mGFP-FTH1 into enlarged endosomes (Fig. S3 A). Collectively, these observations demonstrate that ferritin–NCOA4 condensates are targeted not only by macroautophagy but also by endosomal microautophagy.

### TAX1BP1 is dispensable for the formation of ferritin–NCOA4 condensates but required for their recognition by macroautophagy and microautophagy

We further examined whether TAX1BP1, an adaptor protein required for ferritin turnover (Goodwin et al., 2017), was involved in ferritin–NCOA4 condensate formation. Fluorescence microscopy showed that mGFP-TAX1BP1 colocalized with mRuby3-FTH1 puncta in WT and *FIP200* KO cells (Fig. 5 A), suggesting that TAX1BP1 is also a component of ferritin–NCOA4 condensates. These FTH1$^+$TAX1BP1$^+$ condensates often associated with FTH1$^−$TAX1BP1$^+$ condensates, which were likely SQSTM1 bodies as we previously observed (Kishi-Itakura et al., 2014). However, knockout of TAX1BP1 (Fig. S1 E) did not impede condensate formation (Fig. 5 B). These results denote that

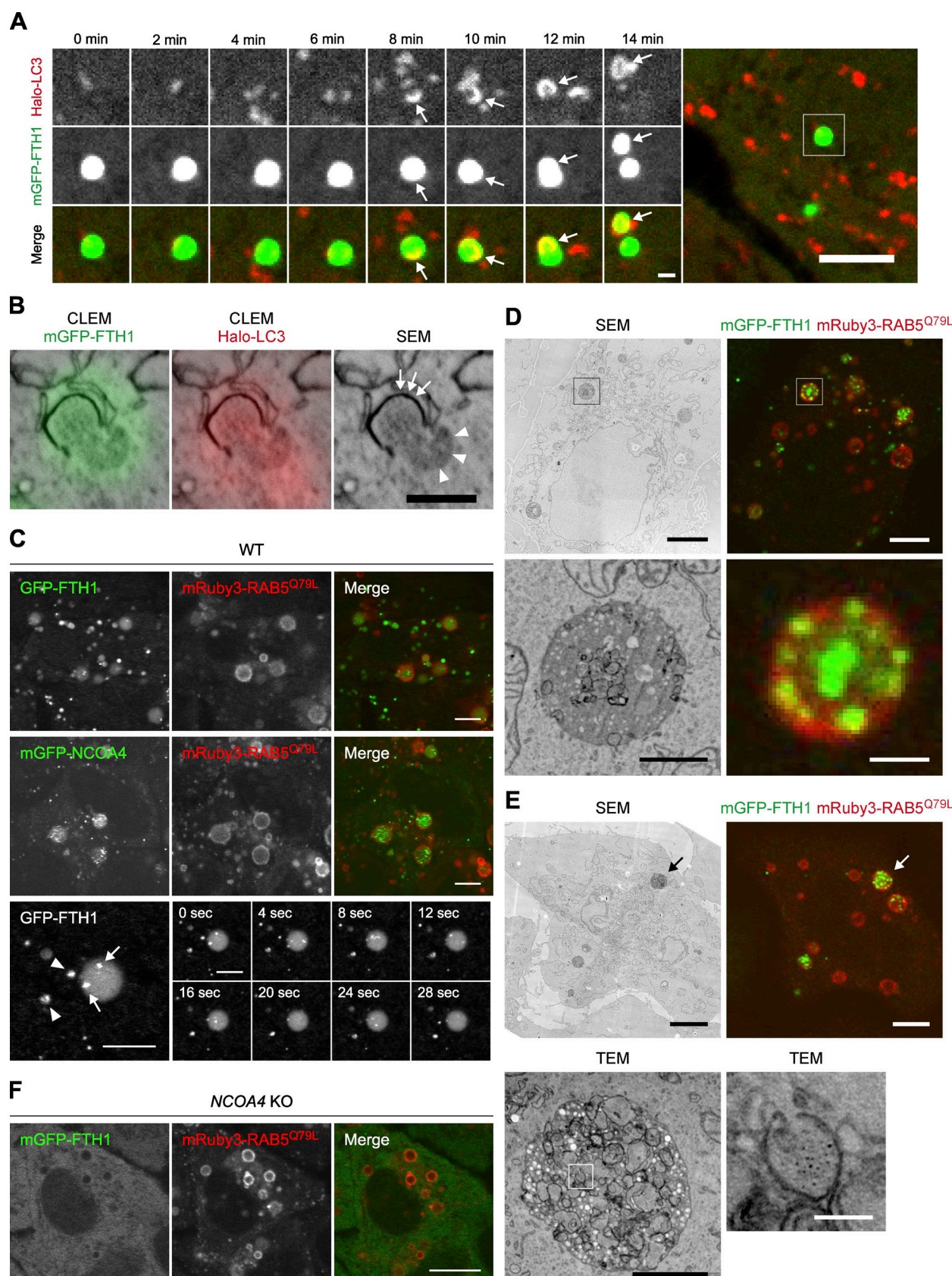

Figure 4. **The ferritin–NCOA4 condensates are targeted by both macroautophagy and endosomal microautophagy. (A)** Ferritin–NCOA4 condensates are engulfed by autophagosomes. WT HeLa cells expressing mGFP-FTH1 (green) and Halo-LC3 (red) were treated with 50 µg/ml FAC for 24 h followed by 50 µM DFO for 5 h and then observed by time-lapse fluorescence microscopy at 2-min intervals (see also Video 1). Arrows indicate part of ferritin condensates

engulfed by an autophagosome. Scale bars, 5 µm (main) and 1 µm (magnified). **(B)** 3D-CLEM of cells treated as in A. Arrowheads indicate the surface of a ferritin–NCOA4 condensate exposed to the cytosol, and arrows indicate an elongating autophagosomal membrane. Scale bar, 500 nm. **(C)** The ferritin–NCOA4 condensates are incorporated into endosomes. WT HeLa cells expressing GFP-FTH1 or mGFP-NCOA4 were treated with 2 µg/ml doxycycline for 48 h to induce mRuby3-RAB5^Q79L expression and then observed by time-lapse fluorescence microscopy at 4-s intervals under normal growing conditions (see also Videos 2, 3, and 4). Arrowheads indicate ferritin–NCOA4 condensates in the cytosol, and arrows indicate the ferritin–NCOA4 condensates in enlarged endosomes. Scale bars, 5 µm. **(D and E)** 3D-CLEM of WT HeLa cells expressing mGFP-FTH1 treated as in C. Correlative scanning electron microscopy (SEM) and fluorescent images are shown. An enlarged endosome containing mGFP-FTH1 puncta is magnified. Scale bars, 5 µm (main) and 1 µm (magnified; D). TEM images of the enlarged endosome indicated by arrows in SEM and fluorescent images are shown. Scale bars, 5 µm (SEM and fluorescent images), 1 µm (TEM), and 100 nm (magnified TEM; E). **(F)** *NCOA4* KO HeLa cells expressing mGFP-FTH1 were examined as in C. Scale bar, 10 µm.

TAX1BP1 interacts with component(s) of ferritin–NCOA4 condensates, probably with NCOA4 (Goodwin et al., 2017), but is dispensable for the formation of ferritin condensates. Given that ferritin turnover was blocked in the absence of TAX1BP1, as reported previously (Goodwin et al., 2017; Fig. S1 F), we assumed that TAX1BP1 functions after the formation of condensates, more specifically, at the recognition step of ferritin–NCOA4 condensates by macroautophagy and/or microautophagy. In WT cells under iron-depleted conditions, Halo-LC3 frequently colocalized with mGFP-FTH1 puncta (Fig. 5, C and D). By contrast, *TAX1BP1* KO cells did not show colocalization of Halo-LC3 with mGFP-FTH1 puncta (Fig. 5, C and D), suggesting that TAX1BP1 is required for the recognition of ferritin–NCOA4 condensates by macroautophagy. Likewise, mGFP-FTH1 puncta were frequently trapped in endosomes in WT cells, but not in *TAX1BP1* KO cells (Fig. 5, E and F).

To further examine the delivery of ferritin-NCOA4 condensates to lysosomes, we used a Halo-NCOA4 reporter because HaloTag becomes stable in lysosomes after ligand binding (Rudinskiy et al., 2022; Yim et al., 2022). In WT cells, signals of Halo-NCOA4 labeled with its fluorescent ligand for 24 h well colocalized with LysoTracker signals, likely resulting from both macroautophagy and microautophagy (Fig. S3, B and C). To specifically track the microautophagy pathway, we used macroautophagy-deficient *FIP200* KO cells and found that colocalization between Halo-NCOA4 and LysoTracker Red was reduced compared with WT cells (Fig. S3, B and C). However, there remained some colocalization signals, which should represent the result of ferritin microautophagy. In contrast, *TAX1BP1* KO cells did not show Halo-NCOA4 colocalization with LysoTracker signals at all. Taken together, we concluded that TAX1BP1 is dispensable for ferritin–NCOA4 condensate formation but is required for the recognition of ferritin condensates as a common adaptor for macroautophagy and endosomal microautophagy.

## Conclusion

In this study, we revealed that ferritin particles undergo phase separation in the cytosol and that NCOA4 functions as a driver of ferritin phase separation by providing multivalent interactions (i.e., homodimerization and the direct binding to FTH1). We also showed that resultant ferritin–NCOA4 condensates were eventually sorted into two different pathways, macroautophagy and endosomal microautophagy, in a TAX1BP1-dependent manner (Fig. 5 G). After we submitted this manuscript, Iwai's group published an independent study showing that NCOA4 forms condensates with fluidity in an iron-dependent manner, and the

iron-induced NCOA4 condensation is required for ATG7-independent, but TAX1BP1-dependent ferritin degradation under prolonged iron-replete conditions (Kuno et al., 2022).

Increasing reports have pointed out the relationship between macroautophagy and phase separation of autophagic cargos such as SQSTM1 bodies (Sun et al., 2018), Ape1 condensates (Yamasaki et al., 2020), and PGL granules (Zhang et al., 2018). Autophagosomal sequestration of liquid droplets is promoted by the wetting effect resulting from contact between liquid droplets and membranes (Agudo-Canalejo et al., 2021). This process was termed "fluidophagy," highlighting the importance of the liquidity of droplets in the deformation of the autophagosomal membranes. Since the autophagosomal membranes adhered to the surface of liquid-like ferritin–NCOA4 condensates (Fig. 4 B), we propose that ferritin macroautophagy (macroferritinophagy) is a type of "macrofluidophagy." In addition, our observation that ferritin–NCOA4 condensates were incorporated into endosomes suggests that endosomal microautophagy of ferritin condensates (microferritinophagy) can be regarded as "microfluidophagy," which is also promoted by the wetting effect. If this is the case, LLPS might be a common mechanism for the two distinct lysosomal ferritin transport pathways. It is also well known that ferritin is secreted, and this secretion is enhanced in some diseases, including hemophagocytic lymphohistiocytosis and adult-onset Still's disease (Rosario et al., 2013). NCOA4 and ferritin can be secreted via extracellular vesicles (Solvik et al., 2021; Yanatori et al., 2021). Thus, ferritin–NCOA4 condensates may be directed to both lysosomal degradation and secretion after incorporation into endosomes.

TAX1BP1 has a non-canonical LC3-interacting region (Newman et al., 2012; Tumbarello et al., 2015) and binds to LC3 family proteins, which play a central role in selective macroautophagy (Birgisdottir et al., 2013; Johansen and Lamark, 2020). Thus, the association of ferritin–NCOA4 condensates with autophagosomes can be mediated by the direct binding of TAX1BP1 to LC3 family proteins. However, the mechanism by which TAX1BP1 links ferritin–NCOA4 condensates to endosomal microautophagy has yet to be elucidated. This pathway may not require LC3 binding, given that lysosomal degradation of NCOA4 is partially independent of the LC3 lipidation machinery (Goodwin et al., 2017; Mejlvang et al., 2018). An as yet unidentified factor might be involved in the endosomal microautophagy of ferritin–NCOA4 condensates.

Ferritin clusters have been observed in several types of cells under physiological conditions, including reticulocytes (Heynen and Verwilghen, 1982; Sullivan et al., 1976), Caco-2 cells (Meyron-Holtz et al., 2014), and human kidney proximal tubule

**Figure 5. TAX1BP1 is dispensable for ferritin–NCOA4 condensate formation but required for its recognition for macroautophagy and endosomal microautophagy. (A)** Colocalization of TAX1BP1 with ferritin condensates. WT and *FIP200* KO HeLa cells expressing mGFP-TAX1BP1 and mRuby3-FTH1 were grown in DMEM and observed by fluorescence microscopy. Scale bars, 10 μm (main) and 1 μm (inset). **(B)** TAX1BP1 is dispensable for condensate formation. *TAX1BP1* KO and *FIP200 TAX1BP1* DKO HeLa cells expressing mGFP-FTH1 were observed as in A. Scale bars, 10 μm (main) and 1 μm (inset). **(C)** Ferritin–NCOA4 condensates are recognized by autophagosomes in a TAX1BP1-dependent manner. WT and *TAX1BP1* KO HeLa cells expressing mGFP-FTH1 (green) and Halo-LC3 (red) were treated with 100 μg/ml FAC for 24 h followed by 50 μM DFO for 5.5 h and then observed by fluorescence microscopy. Scale bars, 5 μm (main)

and 1 µm (inset). **(D)** The colocalization rate of mGFP-FTH1 puncta with Halo-LC3 in C was quantified (*n* = 1,130–2,716). Solid bars indicate the means, and dots indicate the data from three independent experiments. Differences were statistically analyzed by Welch's *t* test (two-tailed test). Data distribution was assumed to be normal, which was not formally tested. **(E)** Ferritin–NCOA4 condensates are incorporated into endosomes in a TAX1BP1-dependent manner. mRuby3-RAB5$^{Q79L}$ was expressed by treatment with 2 µg/ml doxycycline for 48 h in WT and *TAX1BP1* KO HeLa cells expressing mGFP-FTH1. Scale bars, 10 µm. **(F)** The rate of the endosomes containing mGFP-FTH1 puncta in E was quantified (*n* = 359–436). Solid bars indicate the means, and dots indicate the data from three independent experiments. Differences were statistically analyzed by Welch's *t* test (two-tailed test). Data distribution was assumed to be normal, which was not formally tested. **(G)** A model of NCOA4-dependent formation of ferritin condensates and TAX1BP1-dependent recognition of condensates by macroautophagy and endosomal microautophagy.

brush border cells (Cohen et al., 2010), as well as in some disease conditions such as in erythroblasts in sideroblastic anemia (Ghadially, 1975) and in hepatocytes in hemochromatosis (Iancu, 1992). However, it is unknown whether these structures are indeed ferritin–NCOA4 condensates and are degraded by macroautophagy and/or endosomal microautophagy. Further studies will be required to reveal which molecules exist and what kind of reactions occur in ferritin–NCOA4 condensates so that we can gain an understanding of their exact role in iron metabolism.

## Materials and methods
### Cell lines and culture conditions
HeLa and HEK293T cells (authenticated by RIKEN) were cultured in Dulbecco's modified Eagle medium (DMEM; D6546; Sigma-Aldrich) supplemented with 10% fetal bovine serum (FBS; 173012; Sigma-Aldrich) and 2 mM L-glutamine (25030-081; GIBCO) in a 5% $CO_2$ incubator at 37°C. For the iron-replete conditions, HeLa cells were treated with 10, 50, or 100 µg/ml ferric ammonium citrate (FAC; F5879; Sigma-Aldrich). For iron-deficient conditions, cells were treated with FAC for 24 h followed by 50 µM deferoxamine (DFO; D9533; Sigma-Aldrich).

*NCOA4* KO, *FIP200 NCOA4* DKO, *TAX1BP1* KO, and *FIP200 TAX1BP1* DKO HeLa cells were generated as follows: DNA fragments encoding the neomycin-resistant cassette flanked by 500-bp sequences homologous to exon 4 and exon 6 of the *NCOA4* gene or homologous to exon 4 of the *TAX1BP1* gene were prepared. WT and *FIP200* KO (Tsuboyama et al., 2016) HeLa cells were cotransfected with the DNA fragment and the PX459 (#48139; Addgene)-based plasmid expressing Cas9 and gRNA (5′-GTCTTAGAAGCCGTGAGGTA-3′ for the *NCOA4* gene) or gRNA (5′-GTTCTGTTACGTTACCCATA-3′ for the *TAX1BP1* gene) using FuGENE HD (E2311; Promega) for 4 h. The cells were cultured for 5 d in DMEM, treated with 1.5 mg/ml G418 (09380-86; Nacalai Tesque) for 1 wk, and the clones were selected. HeLa cells inducibly expressing mRuby3-RAB5$^{Q79L}$ were generated as follows: a DNA fragment encoding mRuby3-RAB5$^{Q79L}$ under the Tet-on promoter with the hygromycin-resistant cassette was flanked by 500-bp sequences homologous to the AAVS1 locus. WT, *NCOA4* KO, and *TAX1BP1* KO HeLa cells were cotransfected with the DNA fragment and the PX459 (#48139; Addgene)-based plasmid expressing Cas9 and gRNA (5′-GGGGCCACTAGGGACAGGAT-3′). The cells were cultured for 5 d, treated with 50 µg/ml hygromycin (10687010; Thermo Fisher Scientific) for 1 wk, and the clones were selected.

### Plasmids
Plasmids for stable expression in HeLa cells were generated as follows: DNA fragments encoding enhanced GFP, monomeric enhanced GFP (mGFP) harboring A206K mutation, mRuby3 (#74252; codon-optimized from Addgene), HaloTag7 (Halo; N2701; Promega), or 3×FLAG were inserted into the retroviral plasmid pMRX-IP (Kitamura et al., 2003; Saitoh et al., 2003) or pMRX-IB (Morita et al., 2018) by the seamless ligation cloning extract (SLiCE) method (Motohashi, 2017). Then, DNA fragments encoding human FTH1 (NCBI accession no. NP_002023.2), FTL (NP_000137.2), NCOA4 (NP_001138734.1, isoform 3), or TAX1BP1 (NP_001073333.1, isoform 2) were inserted into the pMRX-IP-based or pMRX-IB-based plasmids by the SLiCE method.

For expression in yeast cells, pRS316 (ATCC 77145)-based plasmids expressing monomeric ultra-stable GFP (muGFP; Scott et al., 2018) or muGFP-NCOA4 were generated as follows: a DNA fragment encoding muGFP or muGFP-NCOA4 was inserted downstream of the GPD promoter of pRS316-GPDpro by the SLiCE method. pRS314 (ATCC 77143)-based plasmids expressing FTH1 were generated as follows: a DNA fragment encoding FTH1 was inserted downstream of the GPD promoter of pRS314-GPDpro by the SLiCE method.

For the yeast two-hybrid assay, DNA fragments encoding the N-terminal domain (residues 1–182) of NCOA4 (WT, I56E, or L63R) were inserted into the pGADT7 or pGBKT7 vector (630489; Clontech) by the SLiCE method.

Primers used for plasmid construction are listed in Table S1.

### Stable expression in HeLa cells by retrovirus infection
For preparation of the retrovirus solution, HEK293T cells were transfected with the pMRX-IP-based or pMRX-IB-based retroviral plasmid (Kitamura et al., 2003; Saitoh et al., 2003) together with pCG-gag-pol and pCG-VSV-G (a gift from Dr. T. Yasui, National Institutes of Biomedical Innovation, Health and Nutrition, Osaka, Japan) using Lipofectamine 2000 (11668019; Thermo Fisher Scientific) for 4–6 h. After the cells were cultured for 2–3 d in DMEM, the retrovirus-containing medium was harvested and filtered through a 0.45-µm filter unit (Ultrafree-MC; Millipore) and added to HeLa cells with 8 µg/ml polybrene (H9268; Sigma-Aldrich). After the cells were cultured for 1 d, selection was performed with 1–2 µg/ml puromycin (P8833; Sigma-Aldrich) or 2–3 µg/ml blasticidin (022-18713; Fujifilm Wako Pure Chemical Corporation). GFP-, mGFP-, mRuby3-, HaloTag7-, and 3xFLAG-tagged exogenous proteins (except for mRuby3-RAB5$^{Q79L}$) were stably expressed in HeLa cells by retrovirus infection.

## RNA interference

Stealth RNAi siRNAs (Thermo Fisher Scientific) were used for RNA interference. HeLa cells were transfected with siLuc (5′-CGCGGUCGGUAAAGTTGUUCCAUUU-3′), siFTH1 #1 (5′-CCAGAACUACCACCAGGACUCAGAG-3′), siFTH1 #2 (5′-CAUGUCUUACUACUUUGACCGCGAU-3′), siFTH1 #3 (5′-AGUCACUACUGGAACUGCACAAACU-3′), and/or siFTL (5′-GCAAAGUAAUAGGGCUUCUGCCUAA-3′) using lipofectamine RNAiMAX (13778150; Thermo Fisher Scientific) for 4 h. Then, the cells were cultured for 46 h (siFTH1 and siFTH1 siFTL) or 66 h (siLuc and siFTL) in DMEM.

## Preparation of whole cell lysates

HeLa cells were harvested by centrifugation at 3,000 $g$ for 1 min at 4°C and lysed with 0.2% $n$-dodecyl-β-D-maltoside (DDM; 14239-54; Nacalai Tesque) in 25 mM HEPES-KOH pH 7.2, 150 mM NaCl, 2 mM MgSO$_4$, and 1% protease inhibitor cocktail (P8340; Sigma-Aldrich) for 20 min on ice and then treated with 0.1% benzonase (70664; Millipore). The protein concentrations were determined by a microvolume spectrophotometer (NanoDrop One; Thermo Fisher Scientific). Whole cell lysates were mixed with 2×SDS-PAGE sample buffer and boiled at 98°C for 5 min and the protein concentrations were adjusted with 1×SDS-PAGE sample buffer.

## Coimmunoprecipitation

HeLa cells were solubilized with 0.1% DDM in HNE buffer (25 mM HEPES-KOH pH 7.2, 150 mM NaCl, 2 mM EDTA) containing 1% protease inhibitor cocktail (P8340; Sigma-Aldrich) for 20 min on ice and then centrifuged at 17,700 $g$ for 15 min. The supernatants were incubated with anti-DYKDDDDK/FLAG magnetic beads (017-25151; Fujifilm Wako Pure Chemical Corporation) for 3 h at 4°C. The beads were washed three times with HNE buffer containing 0.05% DDM, and bound proteins were eluted with SDS-PAGE sample buffer at 98°C for 5 min.

## Immunoblotting

Immunoblotting was performed using anti-FTH1 (MA5-32244; Thermo Fisher Scientific), anti-FTL (MA5-32755; Thermo Fisher Scientific), anti-NCOA4 (SAB1409837; Sigma-Aldrich), anti-TAX1BP1 (HPA024432; Sigma-Aldrich), anti-FIP200 (17250-1-AP; Proteintech), anti-HSP90 (610419; BD Transduction Laboratories), anti-GFP (A-6455; Thermo Fisher Scientific), and HRP-conjugated anti-DYKDDDDK/FLAG (015-22391; Fujifilm Wako Pure Chemical Corporation) as primary antibodies, and HRP-conjugated anti-rabbit IgG (111-035-144; Jackson ImmunoResearch) and HRP-conjugated anti-mouse IgG (315-035-003; Jackson ImmunoResearch) as secondary antibodies. Proteins were transferred to Immobilon-P PVDF membrane (IPVH00010; Millipore) with Trans-Blot Turbo Transfer System (Bio-Rad Laboratories). SuperSignal West Pico Chemiluminescent Substrate (1856135; Thermo Fisher Scientific) and Immobilon Western Chemiluminescent HRP Substrate (P90715; Millipore) were used to visualize the signals, which were detected by an image analyzer (FUSION SOLO.7S.EDGE; Vilber-Lourmat). Contrast and brightness adjustments were performed using the ImageJ (National

Institutes of Health) or Photoshop CC 2019/2020 (Adobe) software.

## Yeast transformation

For exogenous expression of muGFP-NCOA4 and FTH1 in yeast cells, BJ2168 (ATCC-208277) $atg11\Delta$ $atg17\Delta$ cells (a gift from Dr. Y. Ohsumi, Tokyo Institute of Technology, Tokyo, Japan) were transformed with the plasmid pRS316-muGFP or pRS316-muGFP-NCOA4 by the high-efficiency yeast transformation method (Gietz and Schiestl, 2007); yeast cells were washed once with 100 mM LiAc and then incubated with the pRS316-based plasmid in 33% (w/v) polyethylene glycol 3350 (202444; Sigma-Aldrich), 100 mM LiAc, and 0.28 mg/ml single-stranded carrier DNA (D9156; Sigma-Aldrich) for 20 min at 42°C and then for 30 min at 30°C. The cells were harvested and grown on SD (-Ura) plates at 30°C. The cells were further transformed with pRS314 or pRS314-FTH1 and grown on SD (-Ura, -Trp) plates. For the yeast two-hybrid assay, AH109 cells were transformed with the pGBKT7-based plasmid and grown on SD (-Trp) plates. The cells were further transformed with the pGADT7-based plasmid and grown on SD (-Trp, -Leu) plates.

## Fluorescence microscopy

Live-imaging fluorescence microscopy was performed using the FV3000 confocal laser microscope (Olympus) equipped with a 60× oil-immersion objective lens (NA 1.4, PLAPON60XOSC2; Olympus) and a stage top CO$_2$ incubator (STXG-IX3WX; Tokai Hit) at 37°C with 5% CO$_2$. HeLa cells were grown in DMEM in a glass-bottom dish (3910-035; Iwaki) and fluorescent images were captured using the FluoView software (Olympus). For observation of Halo-LC3, HeLa cells were incubated with 20 nM HaloTag SaraFluor 650T ligand (GCKA308; Promega) for 15 min before observation. The numbers of punctate structures were counted using Fiji software (National Institutes of Health; Schindelin et al., 2012).

## Fluorescence recovery after photobleaching (FRAP)

FRAP experiments were performed using the FV3000 confocal laser microscope system (Olympus) at 37°C with 5% CO$_2$. Photobleaching of mGFP-FTH1 was achieved using a 488-nm laser with a bleaching time of 55.461 ms. Images were captured at 10-s intervals for 20 min (120 time points) or 4-s intervals for 128 s (32 time points).

## Immunostaining

Immunostaining was performed using anti-FTL (MA5-14733; Thermo Fisher Scientific) and anti-GFP monoclonal antibodies (D153-3; MBL) as primary antibodies, and Alexa Fluor 488-conjugated anti-mouse IgG (A-11001; Thermo Fisher Scientific), Alexa Fluor 488-conjugated anti-rat IgG (A-11006; Thermo Fisher Scientific), and Alexa Fluor 568-conjugated anti-mouse IgG (A-11004; Thermo Fisher Scientific) antibodies as secondary antibodies. HeLa cells grown on a cover glass were washed twice with phosphate-buffered saline (PBS), treated with 4% paraformaldehyde (09154-85; Nacalai Tesque) for 10 min at room temperature, washed three times with PBS, treated with 50 µg/ml digitonin for 5 min at room temperature, washed three

times with PBS, and treated with 3% BSA in PBS for 30 min at room temperature. The cells were incubated with primary antibodies in PBS containing 3% BSA for 1 h at room temperature, washed five times with PBS, incubated with second antibodies in PBS containing 3% BSA for 1 h at room temperature, washed five times with PBS, and then treated with SlowFade Gold Antifade Mountant with DAPI (S36939; Thermo Fisher Scientific).

### Transmission electron microscopy (TEM)
HeLa cells were grown on a Celltight C-1 Celldesk LF coverslip (MS-0113K; Sumitomo Bakelite) in DMEM and fixed with 2.5% glutaraldehyde (G018/1; TAAB) in 0.1 M cacodylate buffer pH 7.4 (37237-35; Nacalai Tesque) for 2 h on ice. After several washes with 0.1 M phosphate buffer, cells were postfixed with 1% osmium tetroxide, dehydrated with ethanol series and propylene oxide, and then embedded in EPON812.

### 3D-CLEM
For observation of the ferritin condensates engulfed by the autophagosomes, HeLa cells expressing mGFP-FTH1 and Halo-LC3 were grown on a glass-bottom dish with 150-μm grids (TCI-3922-035R-1CS; Iwaki, a custom-made product with cover glass attached in the opposite direction) coated with carbon by vacuum evaporator (IB-29510VET) and 0.1% gelatin. The cells were treated with 20 nM HaloTag SaraFluor 650T ligand (GCKA308; Promega) for 15 min before fixation. For observation of the ferritin condensates incorporated into the endosomes or autophagosomes, HeLa cells expressing mGFP-FTH1 were grown as described above and treated with 2 μg/ml doxycycline for 48 h before fixation. The cells were fixed and observed by the FV3000 confocal laser microscope system (Olympus), and then postfixed overnight with 2.5% glutaraldehyde in 0.1 M sodium cacodylate buffer at 4°C. After washing with 0.1 M sodium cacodylate, cells were treated with ferrocyanide-reduced osmium tetroxide (1% $OsO_4$, 1.5% $K_4[Fe(CN)_6]$, 0.065 M sodium cacodylate buffer) for 2 h at 4°C, and rinsed five times using Milli-Q water. The samples were en block stained with 2% uranyl acetate solution then rinsed five times using Milli-Q water. The samples were dehydrated with ethanol series, and then embedded in EPON812. After polymerization, cover glasses were removed by soaking in liquid nitrogen. Then the sample blocks were trimmed to almost the same size as the fluorescent images (about 150 × 150 μm). Serial sections (25 nm thick) were generated using an ultramicrotome (UC7; Leica) and a diamond knife with a large boat (Ultrajumbo, 35° or 45°, Diatome). Sections were collected on a precleaned silicon wafer strip, and then observed under a scanning electron microscope (SEM; JSM7900F; JEOL) and TEM (JEM-1010; JEOL). CLEM images were constructed using Fiji (National Institutes of Health; Schindelin et al., 2012) and Photoshop CC 2019/2020 (Adobe) software.

### Online supplemental material
Fig. S1 shows that exogenously expressed mGFP-FTH1 colocalized with endogenous FTL, and that both mGFP-FTH1 and mGFP-FTL puncta showed fluorescence recovery after photobleaching. Fig. S2 shows a predicted structure of the N-terminal domain of NCOA4. Fig. S3 shows microautophagy of ferritin–NCOA4 condensates in cells deficient for NCOA4 or TAX1BP1. Video 1 shows an mGFP-FTH1 punctum engulfed by an autophagosome. Video 2 shows GFP-FTH1 puncta incorporated into enlarged endosomes. Video 3 shows mGFP-NCOA4 puncta incorporated into enlarged endosomes. Video 4 shows GFP-FTH1 puncta observed in the cytosol and an enlarged endosome. Table S1 lists primers used for plasmid construction.

## Acknowledgments
We are grateful to Maya Shirakawa for technical assistance with the cell biology experiments; Satoru Takahashi, Yoko Ishida, and Keiko Igarashi for technical assistance with the 3D-CLEM experiments; Shoji Yamaoka (Tokyo Medical and Dental University, Tokyo, Japan) for the pMRXIP plasmid; and Teruhito Yasui for the pCG-gag-pol and pCG-VSV-G plasmids.

This work was supported by the Exploratory Research for Advanced Technology (ERATO) research funding program of the Japan Science and Technology Agency (JST; JPMJER1702 to N. Mizushima) and a Grant-in-Aid for Transformative Research Areas (A) (21H05256 to H. Yamamoto) and for Specially Promoted Research (22H04919 to N. Mizushima) from the Japan Society for the Promotion of Science (JSPS).

The authors declare no competing financial interests.

Author contributions: T. Ohshima, H. Yamamoto, and N. Mizushima designed the project. T. Ohshima and H. Yamamoto performed the cell biology experiments. Y. Sakamaki and C. Saito performed the TEM and 3D-CLEM experiments. T. Ohshima, H. Yamamoto, and N. Mizushima wrote the manuscript. All authors commented on the manuscript.

Submitted: 30 March 2022

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

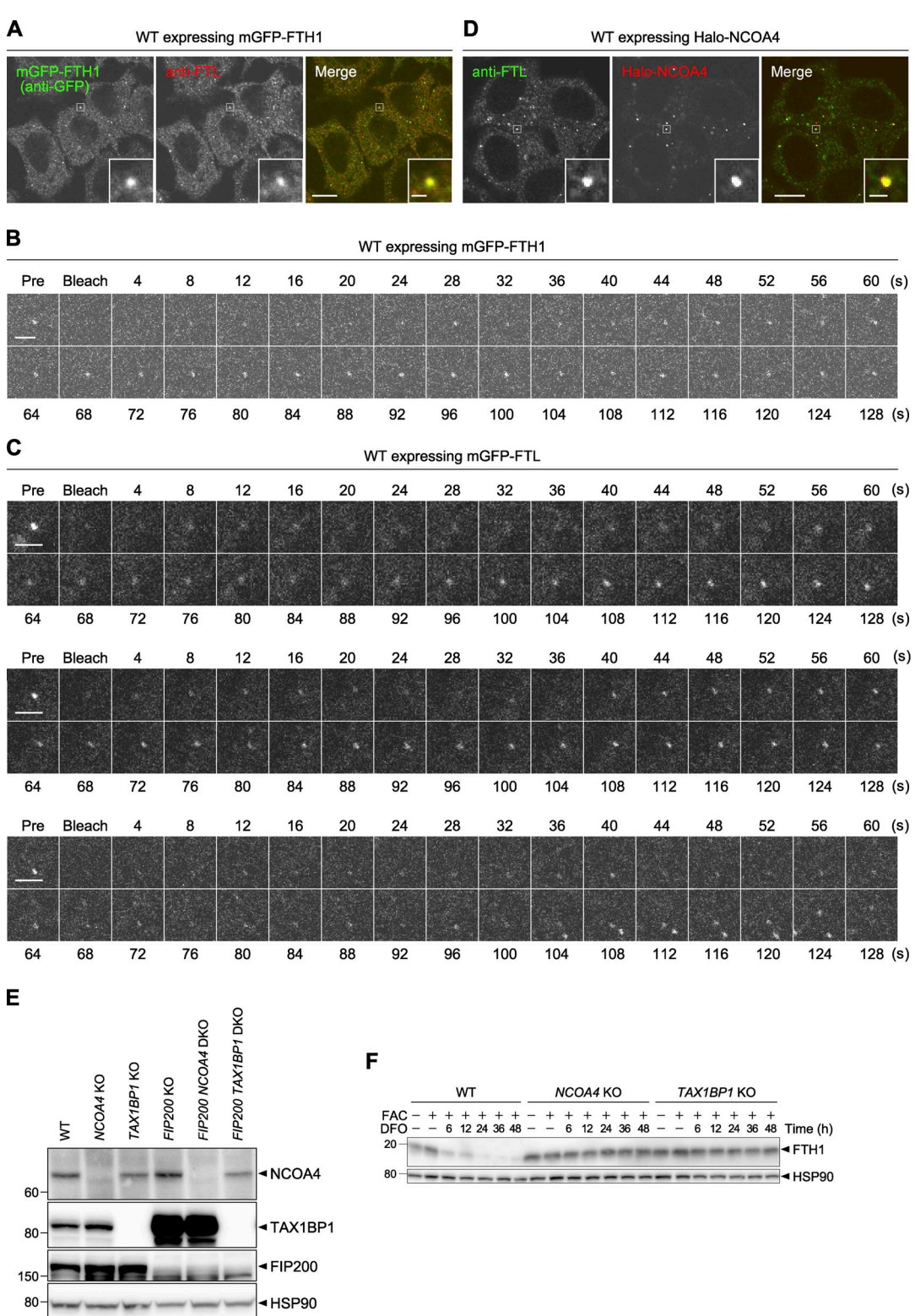

**Figure S1.** **Ferritin forms liquid-like condensates with NCOA4. (A and D)** WT HeLa cells expressing mGFP-FTH1 (A) or Halo-NCOA4 (D) were grown in DMEM, treated with 100 nM HaloTag SaraFluor 650T ligand (D), and then subjected to immunostaining with anti-FTL monoclonal antibody. Scale bars, 10 µm (main) and 1 µm (inset). **(B and C)** FRAP analyses of ferritin condensates. WT HeLa cells expressing mGFP-FTH1 (B) or mGFP-FTL (C) were treated with 100 µg/ml FAC for 24 h followed by 50 µM DFO for 2 h, and subjected to FRAP analyses (4-s intervals for 128 s). Three individual mGFP-FTL puncta are shown in C. Scale bars, 2 µm. **(E)** WT, *NCOA4* KO, *TAX1BP1* KO, *FIP200* KO, *FIP200 NCOA4* DKO, and *FIP200 TAX1BP1* DKO HeLa cells were grown in DMEM. Whole-cell lysates were analyzed by immunoblotting with antibodies against NCOA4, TAX1BP1, FIP200, and HSP90. **(F)** WT, *NCOA4* KO, and *TAX1BP1* KO cells were grown in DMEM and treated with 10 µg/ml FAC for 24 h followed by 50 µM DFO for the indicated hours. Whole-cell lysates were analyzed by immunoblotting with antibodies against FTH1 and HSP90. Source data are available for this figure: SourceData FS1.

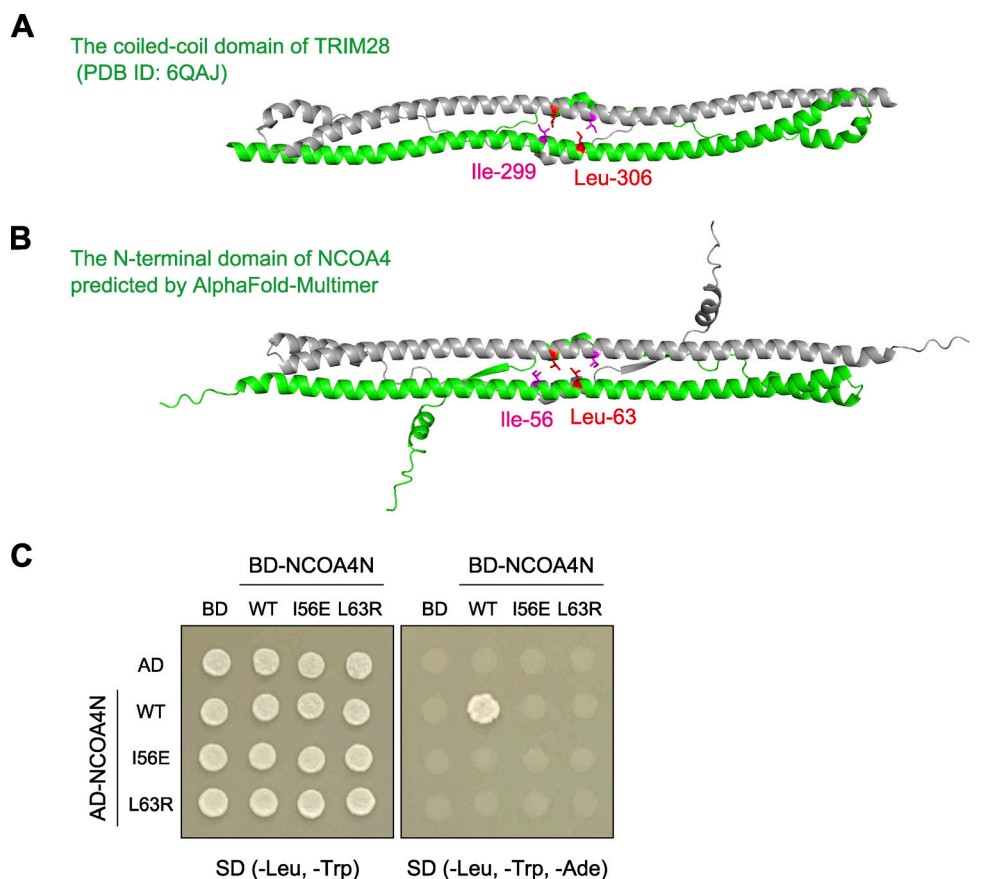

**A** The coiled-coil domain of TRIM28
(PDB ID: 6QAJ)

Ile-299   Leu-306

**B** The N-terminal domain of NCOA4
predicted by AlphaFold-Multimer

Ile-56   Leu-63

**C**

|  | BD-NCOA4N | | | | BD-NCOA4N | | |
|---|---|---|---|---|---|---|---|
|  | BD | WT | I56E | L63R | BD | WT | I56E | L63R |

AD-NCOA4N: AD, WT, I56E, L63R

SD (-Leu, -Trp)          SD (-Leu, -Trp, -Ade)

Figure S2.   **The N-terminal domain of NCOA4 is predicted to form a homodimer. (A)** An HHpred search showed that the N-terminal domain of NCOA4 is similar to the coiled-coil domain (residues 244–405) of TRIM28, which forms a homodimer. The coiled-coil domains of TRIM28 (green and gray) are shown (PDB accession no. 6QAJ). **(B)** The structure of the N-terminal domain (residues 1–182) of NCOA4 (green and gray) is predicted by AlphaFold-Multimer. The putative self-interaction sites Ile-56 (magenta) and Leu-63 (red) are shown. **(C)** Yeast AH109 cells were transformed with plasmids expressing the N-terminal domain (residues 1–182) of NCOA4 with or without I56E or L63R mutation fused with a transcription activation domain (AD) or a DNA-binding domain (BD). The cells were grown on SD (-Leu, -Trp) or SD (-Leu, -Trp, -Ade) plates.

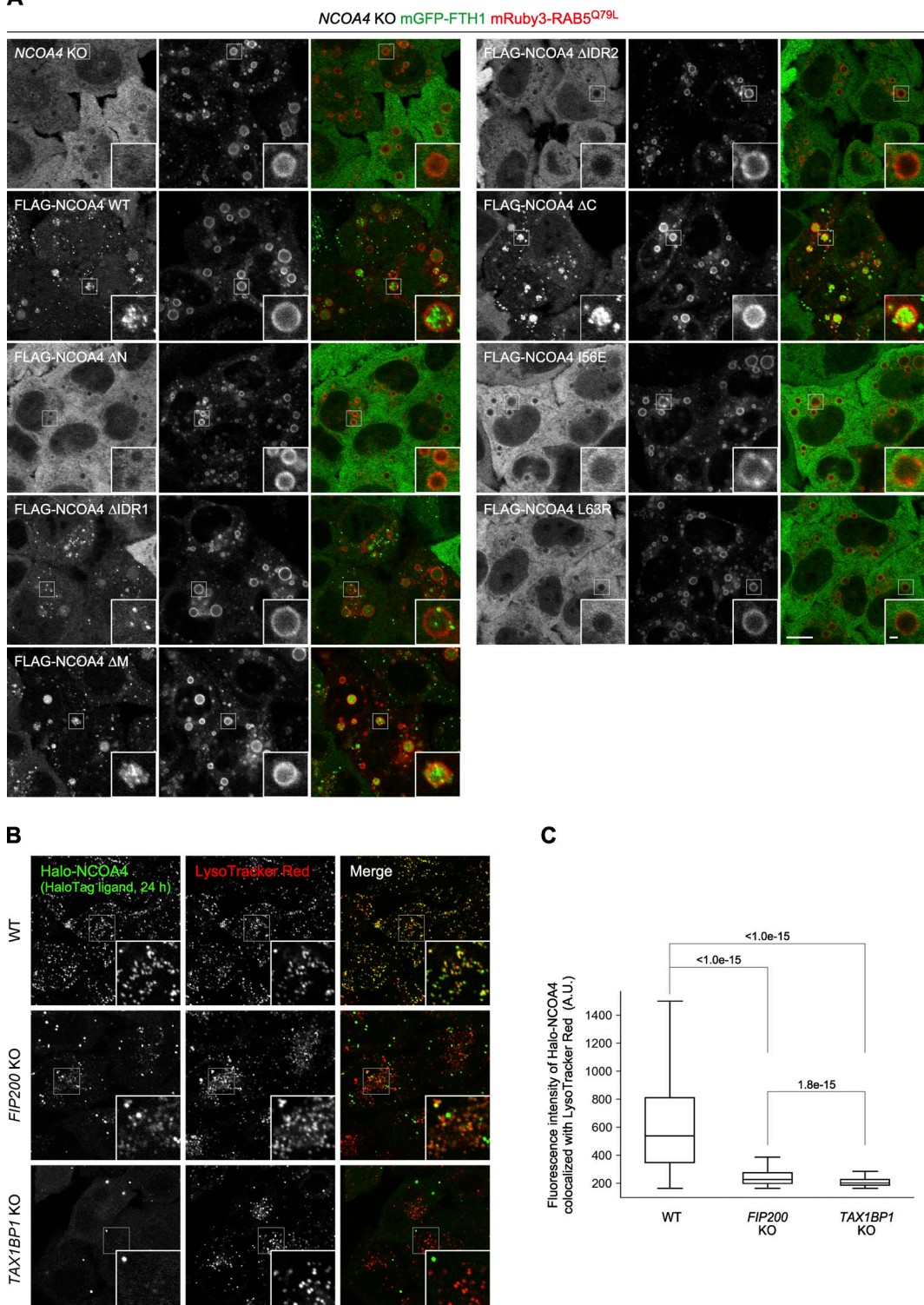

Figure S3.  **The ΔN, ΔIDR2, I56E, and L63R mutants of NCOA4 showed defects in ferritin microautophagy, and TAX1BP1 is required for both macroautophagy and microautophagy of ferritin–NCOA4 condensates. (A)** *NCOA4* KO HeLa cells expressing mGFP-FTH1 and FLAG-NCOA4 (WT or mutants) were treated with 2 μg/ml doxycycline for 30 h to induce mRuby3-RAB5$^{Q79L}$ expression and then observed by fluorescence microscopy. Scale bars, 10 μm (main) and 1 μm (inset). **(B)** WT, *FIP200* KO, and *TAX1BP1* KO HeLa cells expressing Halo-NCOA4 were treated with 20 nM HaloTag SaraFluor 650T ligand for 24 h in DMEM. The cells were further treated with LysoTracker Red for 15 min in DMEM and then observed by fluorescence microscopy. Scale bars, 10 μm (main) and 3 μm (inset). **(C)** Fluorescence intensity of Halo-NCOA4 per LysoTracker Red-positive structure in B was quantified (*n* = 2,235–3,626 LysoTracker Red-positive structures from a subset of observations reproduced in at least two independent experiments). Differences were statistically analyzed by Welch's *t* test with Holm–Sidak method for multiple comparison (two-tailed test). Data distribution was assumed to be normal, which was not formally tested.

Video 1.   **A ferritin–NCOA4 condensate is engulfed by an autophagosome.** Fluorescent images of mGFP-FTH1 (green) and Halo-LC3 (red) in WT HeLa cells (used in Fig. 4 A) were captured by time-lapse fluorescence microscopy at 2-min intervals and are shown at 7 fps.

Video 2.   **Ferritin–NCOA4 condensates labeled with GFP-FTH1 are incorporated into enlarged endosomes.** Fluorescent images of GFP-FTH1 (white) in WT HeLa cells expressing mRuby3-RAB5$^{Q79L}$ (used in Fig. 4 C, upper panels) were captured by time-lapse fluorescence microscopy at 4-s intervals and are shown at 10 fps (mRuby3 is not shown).

Video 3.   **Ferritin–NCOA4 condensates labeled with mGFP-NCOA4 are incorporated into enlarged endosomes.** Fluorescent images of mGFP-NCOA4 (white) in WT HeLa cells expressing mRuby3-RAB5$^{Q79L}$ (used in Fig. 4 C, middle panels) were captured by time-lapse fluorescence microscopy at 4-s intervals and are shown at 10 fps (mRuby3 is not shown).

Video 4.   **Ferritin–NCOA4 condensates labeled with GFP-FTH1 are observed in the cytosol and an enlarged endosome.** Fluorescent images of GFP-FTH1 (white) in WT HeLa cells expressing mRuby3-RAB5$^{Q79L}$ (used in Fig. 4 C, lower panels) were captured by time-lapse fluorescence microscopy at 4-s intervals and are shown at 5 fps (mRuby3 is not shown).

**Provided online is Table S1. Table S1 lists primers used for plasmid construction.**

