## [Peer Review File · The Journal of Cell Biology]

NCOA4 drives ferritin phase separation to facilitate macroferritinophagy and microferritinophagy

Tomoko Ohshima, Hayashi Yamamoto, Yuriko Sakamaki, Chieko Saito, and Noboru Mizushima

Corresponding Author(s): Noboru Mizushima, The University of Tokyo, Graduate School and Faculty of Medicine

Review Timeline:

Submission Date:	2022-03-30
Editorial Decision:	2022-04-08
Revision Received:	2022-07-15
Editorial Decision:	2022-07-19
Revision Received:	2022-07-21

Monitoring Editor: Hong Zhang

Scientific Editor: Dan Simon

Transaction Report:

DOI: <https://doi.org/10.1083/jcb.202203102>

Revision 0

Review #1

1. Evidence, reproducibility and clarity:

Evidence, reproducibility and clarity (Required)

In the manuscript, the authors proposed that ferritin particles undergo phase separation. Ferritin condensate with NCOA4 and can follow two different routes: lysosome-macroautophagys or endosomes-microautophagy. Ferritin-NCOA4-TAX1BP1 and LC3B binding determines the delivery to the lysosomes while the sorting to endosomes seems to be LC3B independent.

The paper is well written and easy to follow. Experiments are properly controlled and the biochemistry, to explain the complexes formation and molecular interaction, is solid. There are no major issues regarding the presented experiments.

To better clarify the dual role of TAX1BP1, authors may consider to investigate the role of p62, NBR1, OPTN in the Ferritin-NCOA4 complex. If any of them is involved in lysosomal or endosomal sorting this may help help to clarify the molecular mechanisms. p62 role in phase separation has been described and more in general these autophagy receptors have several common functions.

****Minor comment:**** check the figure format because some of them have been stretched likely during pdf assembly.

2. Significance:

Significance (Required)

The manuscript provide a relative simple information of great value because it feels a gap in the Ferritin-NCOA4 (Ferritinophagy) field. For sure this finding is of potential interest for the autophagy community and also for researcher devoted to the study of human diseases linked to iron imbalance.

****Field of expertise:****

Autophagy, selective-autophagy

****Referees cross-commenting****

The paper is interesting and the experiments well performed. Addressing the major points should be feasible for the authors.

3. How much time do you estimate the authors will need to complete the suggested revisions:

Estimated time to Complete Revisions (Required)

(Decision Recommendation)

Between 1 and 3 months

4. Review Commons values the work of reviewers and encourages them to get credit for their work. Select 'Yes' below to register your reviewing activity at Publons; note that the content of your review will not be visible on Publons.

Reviewer Publons

Yes

Review #2

1. Evidence, reproducibility and clarity:

Evidence, reproducibility and clarity (Required)

In this study, the authors discover that NCOA4 mediates phase separation of ferritin particles. This function of NCOA4 requires the self interaction of NCOA4 through its N-terminal coiled coil region. Disrupting the self interaction of NCOA4 prevents the phase separation of ferritin particles and blocks ferritin degradation. The findings of this study shed important new light on the functions of NCOA4 in selective autophagy and reveal interesting connections between phase separation and selective autophagy.

I have the following suggestions for the authors to consider.

****Major point:****

In Figure 4F, the authors show that in NCOA4 KO cells, mGFP-FTH1 is no longer incorporated into endosomes. The authors draw the conclusion that "the formation of NCOA4-mediated condensates is required for the incorporation of ferritin into endosomes." I think this conclusion can be strengthened by testing whether mutant forms of NCOA4 can rescue this phenotype.

****Minor points:****

(1) Supplementary Figure S2

The predicted structure shown in the figure is a monomeric structure, perhaps because the software used for the prediction cannot predict homo-oligomeric structures. The AlphaFold-Multimer software can predict homo-oligomeric structures and may be worth trying.

(2) Figure 5G

In the model, the two subunits of an NCOA4 homodimer interact with two ferritin particles rather than interact with FTH1 molecules within a single ferritin particle. The authors may want to describe this particular aspect of the model and discuss the possible underlying mechanism.

(3) page 7, line 189

"these results suggest that condensate formation rather than NCOA4-ferritin binding itself is important for ferritin degradation."

I think a better way to summarize the results may be something like "these results suggest that NCOA4 dimerization-mediated condensate formation is important for ferritin degradation."

(4) page 12, line 335

"kidney proximal tubule brush border cells (Cohen et al.) "

The citation is missing the year of publication.

(5) Figure 3B

The Δ IDR1 panel is missing a small square indicating the region corresponding to the inset.

(6) Abstract

"both require NCOA4, which is thought to be an autophagy receptor for ferritin. However, the exact function of NCOA4 remains elusive."

I suggest rephrasing "the exact function of NCOA4 remains elusive" to something like "it is unclear whether NCOA4 only acts as an autophagy receptor in ferritin degradation".

2. Significance:

Significance (Required)

This study uncovers a previous unknown role of NCOA4 in selective autophagy. The findings reported in this manuscript broaden the understandings of the mechanisms and significances of phase separation in selective autophagy. This paper should be of interest to scientists studying

autophagy and phase separation. My expertise is in selective autophagy.

****Referees cross-commenting****

I think the points raised by the other two reviewers are all valid and worthy of consideration by the authors. It will probably take too long to perform all the additional experiments suggested by the three reviewers. It will be OK with me if the authors decide to only pursue some but not all suggested experiments.

3. How much time do you estimate the authors will need to complete the suggested revisions:

Estimated time to Complete Revisions (Required)

(Decision Recommendation)

Between 1 and 3 months

4. Review Commons values the work of reviewers and encourages them to get credit for their work. Select 'Yes' below to register your reviewing activity at Publons; note that the content of your review will not be visible on Publons.

Reviewer Publons

Yes

Review #3

1. Evidence, reproducibility and clarity:

Evidence, reproducibility and clarity (Required)

The manuscript from Ohshima and colleagues describes liquid-liquid phase separation (LLPS) of ferritin, and its impact on macro- and endosomal micro-autophagy of ferritin particles (namely macro-ferritinophagy and micro-ferritinophagy, respectively). Of note, in this manuscript, endosomal micro-autophagy of ferritin is described for the first time. From a mechanistic viewpoint, the ferritin-cargo receptor NCOA4 drives ferritin LLPS through homo-dimerisation and interaction with FTH1, which is a constituent of ferritin particles. Interestingly, this

phenomenon is functional to ferritin degradation through both macroferritinophagy and microferritinophagy. Also, the degradation of ferritin-NCOA4 condensates occurs in a TAX1BP1-dependent manner, in both macroferritinophagy and microferritinophagy.

Altogether, the study is technically elegant and sound. It relies on exogenous reporters monitored through imaging/live-imaging, several Knock-Out (KO) models, and an elegant NCOA4-FTH1 interaction assay carried out in yeast, which does not express yeast homologues for the proteins of interest. Also, the reported results are of great interest for the field of selective autophagy. However, small implementations are suggested to further strengthen the results. In particular, the data reporting on ferritin being a substrate of endosomal microautophagy should be better related to the current knowledge of the pathway.

****Major comments****

Fig. 1:

- As many other figures along the manuscript, figure 1 relies on the imaging of exogenous reporters: mGFP-FTH1/FTL. While these reporters look reliable on the basis of their localization and changes upon autophagy modulation, a co-staining of mGFP and an endogenous marker of ferritin is suggested, in order to verify the proper localization of the exogenous reporters (i.e. mGFP-FTH1 cells stained with an antibody against endogenous FTL, and mGFP-FTL cells stained with an antibody against endogenous FTH1).
- Fig. 1E-G: this type of experiments are crucial in order to determine the LLPS properties of ferritin. Therefore, I would suggest to perform it also in mGFP-FTL expressing cells, to further strengthen the data already shown for mGFP-FTH1.

Fig. 2:

- Fig. 2C: While the phenotype displayed looks clear, the figure would benefit from a nuclei counterstain, in order to assess whether a comparable number of cells is present in each field.
- Fig. 2D: Please add a loading control for the western blot, in order to assess whether the modulations of the proteins shown are relevant.

Fig. 3:

- Fig. 2F. The FTH1 western blot shows three bands: the mGFP-FTH1 fusion band, the 'endogenous' FTH1 (lower arrowhead), and an even lower band. Can you clarify what the lowest band represents? Of note, it is not clear what band has been quantified by densitometry among the three different ones: please clarify. In relation to the modulation of FTH1 expression, a less exposed western blot would also help appreciating its differences by eye. Since the whole ferritin is supposed to be substrate of macroferritinophagy and microferritinophagy, it would be more solid to rely on both ferritin markers rather than on only one. For this reason, it should be investigated how FTL is modulated upon FAC and DFO treatments, by means western blot analysis and the relative densitometric quantification.
- Fig 2H: For this western blot as well, less exposure time would make the results stand out more

clearly.

Fig. 5:

- The evidence that ferritin-NCOA4 condensates are engulfed into endosomes in a TAX1BP1-dependent manner is very interesting. Does this ability of TAX1BP1 rely on its noncanonical LIR-motif?
- In addition, does ferritin engulfment into endosomes rely on the ESCRT-III system, as observed for other substrates of the endosomal microautophagy (<https://doi.org/10.1083/jcb.201711002>)?
- Also, what is the hypothesis regarding the endpoint of the endosomal route? Are the endosomes delivered to the lysosome, as one would expect if this is endosomal microautophagy. This should be properly tested by co-localisation of ferritin-NCOA4 containing endosomes and lysosomes, for example by using an endogenous marker of the endosomes (e.g. CD63, as previously reported here <https://doi.org/10.1083/jcb.201711002>) and the lysosomal membrane marker LAMP1, in order to strengthen the evidence of ferritin-NCOA4 condensates as substrates for endosomal microautophagy.
- In addition, it would be interesting to assess the contribution of endosomal microautophagy to ferritin degradation both in basal condition and upon induction (i.e. upon FAC followed by DFO treatment, similarly to what was done in fig. 3F). This can be tested by disruption of the multivesicular body dynamics via U18666A treatment, as previously shown (<https://doi.org/10.1083/jcb.201711002>). Along the same line, the authors could investigate whether microferritinophagy still occur upon upstream inhibition of autophagy (e.g. by means of ATG5/7 KO or FIP200 KO, or by treating cells with an ULK1 inhibitor)? In fact, if endosomal microautophagy contributes to ferritin degradation, partial degradation should still be observed under these conditions. Also, microferritinophagy-mediated degradation should be fully abolished upon MVB disruption and/or upon Vps34 inhibition, combined to upstream autophagy inhibition (<https://doi.org/10.1083/jcb.201711002>).

****Minor comments****

Fig. 1:

- Please, clarify in the methods section how the mGFP-FTH1 cells were established. Is the expression of the transgene(s) constitutive or inducible? Similarly, please state the cell type used for each experiment in the relative figure legends.

Fig. 2:

- FTH1, but not FTL, is required for the formation of NCOA4 condensate, knockdown of FTH1 also reduced the expression levels of mGFP-NCOA4, which made it difficult to evaluate the specific role of FTH1 (Figures 2C, 2D). As for the authors' admission, knock-down of FTH1 reduces the expression levels of mGFP-NCOA4, hence preventing from concluding much in regard to the direct involvement of FTH1 in the condensates formation. For this reason, we suggest to stress that the mGFP-NCOA4 reporter is affected by the knockdown of FTH1, with

either oligo interference, and for this reason no clear conclusions can be drawn from this experiment. Alternatively, the authors can either strengthen this experiment (e.g. by means of different approaches or testing different oligo interference against FTH1) or remove it.

- Quantification of the imaging experiments in Fig. 2 should be performed, i.e. number of ferritin-NCOA4 condensates in WT vs. KO cells. Also, the number of cells analyzed should be reported in the figure legend.

Fig. 3:

- Fig. 3C: According to the quantification of Fig. 3B, the effect of introducing the delta IDR1 mutant is equally significant as delta N and delta IDR2. What is the reasoning for stating that the delta IDR1 only results in a partial rescue, while delta N and delta IDR2 cause a full rescue?

2. Significance:

Significance (Required)

Recently, liquid-liquid phase separation (LLPS) has been reported to impact cargo selection in selective autophagy mediated by macroautophagy. At present, nothing is known about the relationship between LLPS, ferritin degradation and endosomal microautophagy. This manuscript describes ferritin to have LLPS properties and to be a substrate of endosomal microautophagy. Hence, this paper fills a gap in our knowledge and suggests that LLPS is relevant to endosomal microautophagy, as well as to ferritin homeostasis.

Of note, increased ferritin secretion is observed in several diseases. As a result, the present study is of potential interest for the clinically relevant features of such diseases.

****Referees cross-commenting****

I agree, the paper quality is altogether high. In the case of my review, I consider the part regarding Figure 1 (expts. on endogenous targets) of the highest priority.

3. How much time do you estimate the authors will need to complete the suggested revisions:

Estimated time to Complete Revisions (Required)

(Decision Recommendation)

Between 1 and 3 months

4. Review Commons values the work of reviewers and encourages them to get credit for their work. Select 'Yes' below to register your reviewing activity at Publons; note

that the content of your review will not be visible on Publons.

Reviewer Publons

Yes

Revision Plan

Manuscript number: RC-2022-01273

Corresponding author(s): Noboru Mizushima

1. General Statements [optional]

We would like to thank all three Reviewers for their positive and constructive comments on our manuscript. We have decided to submit this manuscript as a Report because it contains a sharp message and cutting-edge findings of immediate interest to many fields such as autophagy, endocytosis, liquid-liquid phase separation, and iron metabolisms.

The Reviewers' comments/suggestions are mostly valid. We believe that we can revise our manuscript accordingly. While Reviewer #1 has no major issues, he/she suggests further investigation on the role of p62, NBR1, and OPTN on ferritin turnover to better clarify the dual role of TAX1BP1. Also, Reviewer #3 asked about the mechanism of TAX1BP1 recruitment and the involvement of ESCRTs in his/her major comments #8 and #9. We are now conducting a comprehensive study to reveal the role of different types of autophagy adaptor proteins (including TAX1BP1) in microautophagy. As this topic is rather general and not specific to ferritin-NCOA4 condensates, we think that it would be beyond the scope of this short Report article focusing on ferritin-NCOA4 condensate formation. We would like to prepare it as another full story. We will explain this in Section 4 below. In fact, during Referee cross-commenting, Reviewer #2 said "It will probably take too long to perform all the additional experiments suggested by the three reviewers. It will be OK with me if the authors decide to only pursue some but not all suggested experiments" and Reviewer #3 said "I agree, the paper quality is altogether high. In the case of my review, I consider the part regarding Figure 1 (expts. on endogenous targets) of the highest priority".

We can address all the other comments either experimentally or texturally.

2. Description of the planned revisions

Reviewer #2, Major point

In Figure 4F, the authors show that in NCOA4 KO cells, mGFP-FTH1 is no longer incorporated into endosomes. The authors draw the conclusion that “the formation of NCOA4-mediated condensates is required for the incorporation of ferritin into endosomes.” I think this conclusion can be strengthened by testing whether mutant forms of NCOA4 can rescue this phenotype.

Response:

This is a good idea to further confirm the necessity of NCOA4-mediated phase separation in this process. Now, we are establishing NCOA4 KO HeLa cells expressing the NCOA4 truncation mutants with mGFP-FTH1 and mRuby3-Rab5^{Q79L} and will include the results by fluorescence microscopy as in Figure 4F.

Reviewer #2, Minor point (2)

Figure 5G

In the model, the two subunits of an NCOA4 homodimer interact with two ferritin particles rather than interact with FTH1 molecules within a single ferritin particle. The authors may want to describe this particular aspect of the model and discuss the possible underlying mechanism.

Response:

Both intraparticle and interparticle interactions between an NCOA4 homodimer and FTH1 should be possible. We drew only the latter in the scheme for simplicity, but it might be confusing. To avoid this confusion, we will add the former type of interaction in the model in Figure 5G and discuss the intra- and inter-molecular interactions in the *Discussion* part.

Reviewer #3, Major comment (1)

Fig. 1: As many other figures along the manuscript, figure 1 relies on the imaging of exogenous reporters: mGFP-FTH1/FTL. While these reporters look reliable on the basis of their localization and changes upon autophagy modulation, a co-staining of mGFP and an endogenous marker of ferritin is suggested, in order to verify the proper localization of the exogenous reporters (i.e. mGFP-FTH1 cells stained with an antibody against endogenous FTL, and mGFP-FTL cells stained with an antibody against endogenous FTH1).

Response:

We also appreciate the importance of the proper localization of endogenous ferritin as Reviewer #3 pointed out. We will confirm colocalization between endogenous FTL/FTH1 and exogenous mGFP-FTH1/FTL. If immunostaining does not work well, we

Revision Plan

will biochemically investigate whether exogenous mGFP-FTH1/FTL forms 24-mer ferritin particles with endogenous FTH1 and FTL, which ensures the proper localization of the exogenous reporters mGFP-FTH1 and mGFP-FTL.

Reviewer #3, Major comment (2)

Fig. 1E-G: This type of experiments is crucial in order to determine the LLPS properties of ferritin. Therefore, I would suggest to perform it also in mGFP-FTL expressing cells, to further strengthen the data already shown for mGFP-FTH1.

Response:

We have already tried the FRAP experiment using mGFP-FTL-expressing HeLa cells, but the mGFP-FTL puncta were small and moved fast, making it difficult to track and capture the fluorescent signals after photobleach. This is probably because FTH is more important to generate large condensates. However, we will try the FRAP experiment again for mGFP-FTL.

Reviewer #3, Major comment (6)

Since the whole ferritin is supposed to be substrate of macroferritinophagy and microferritinophagy, it would be more solid to rely on both ferritin markers rather than on only one. For this reason, it should be investigated how FTL is modulated upon FAC and DFO treatments, by means western blot analysis and the relative densitometric quantification.

Response:

We will repeat the western blot analysis of FTL and include the results.

Reviewer #3, Major comment (10)

Also, what is the hypothesis regarding the endpoint of the endosomal route? Are the endosomes delivered to the lysosome, as one would expect if this is endosomal microautophagy. This should be properly tested by co-localisation of ferritin-NCOA4 containing endosomes and lysosomes, for example by using an endogenous marker of the endosomes (e.g. CD63, as previously reported here <https://doi.org/10.1083/jcb.201711002>) and the lysosomal membrane marker LAMP1, in order to strengthen the evidence of ferritin-NCOA4 condensates as substrates for endosomal microautophagy.

Response:

We will investigate the fate of ferritin-NCOA4 condensates incorporated in endosomes by using endosomal and lysosomal markers. We may use ATG KO cells (e.g., *ATG3* KO or *ATG5* KO cells) to exclude the effect of macroautophagy.

Revision Plan

Reviewer #3, Major comment (11)

In addition, it would be interesting to assess the contribution of endosomal microautophagy to ferritin degradation both in basal condition and upon induction (i.e. upon FAC followed by DFO treatment, similarly to what was done in fig. 3F). This can be tested by disruption of the multivesicular body dynamics via U18666A treatment, as previously shown (<https://doi.org/10.1083/jcb.201711002>).

Response:

This is an interesting experiment. We will perform the suggested experiments to examine the contribution of endosomal microautophagy to ferritin degradation using the cholesterol transport inhibitor U18666A.

Reviewer #3, Major comment (12)

Along the same line, the authors could investigate whether microferritinophagy still occur upon upstream inhibition of autophagy (e.g. by means of ATG5/7 KO or FIP200 KO, or by treating cells with an ULK1 inhibitor)? In fact, if endosomal microautophagy contributes to ferritin degradation, partial degradation should still be observed under these conditions. Also, microferritinophagy-mediated degradation should be fully abolished upon MVB disruption and/or upon Vps34 inhibition, combined to upstream autophagy inhibition (<https://doi.org/10.1083/jcb.201711002>).

Response:

We will perform the suggested experiments with the combination of U18666A treatment (the MVB inhibitor) and upstream inhibition of autophagy (*FIP200* KO, *ATG5* KO, etc.) or MRT68921 treatment (the ULK1 inhibitor).

Reviewer #3, Minor comment (3)

Quantification of the imaging experiments in Fig. 2 should be performed, i.e. number of ferritin-NCOA4 condensates in WT vs. KO cells. Also, the number of cells analyzed should be reported in the figure legend.

Response:

We will quantify the results and statistically analyze them.

Revision Plan

3. Description of the revisions that have already been incorporated in the transferred manuscript

Reviewer #1, Minor comment

check the figure format because some of them have been stretched likely during pdf assembly.

Response:

We have checked the figure format and corrected the figures.

Reviewer #2, Minor point (1)

Supplementary Figure S2

The predicted structure shown in the figure is a monomeric structure, perhaps because the software used for the prediction cannot predict homo-oligomeric structures. The AlphaFold-Multimer software can predict homo-oligomeric structures and may be worth trying.

Response:

This is a helpful suggestion. Using the AlphaFold-Multimer software, we succeeded in predicting the structure of the homodimer of the N-terminal domain of NCOA4 as shown below. We have revised the text (Lines 167–168) and Supplementary Figure S2B.

Reviewer #2, Minor point (3)

page 7, line 191

“these results suggest that condensate formation rather than NCOA4-ferritin binding itself is important for ferritin degradation.” I think a better way to summarize the results may be something like “these results suggest that NCOA4 dimerization-mediated condensate formation is important for ferritin degradation.”

Response:

We have changed the text accordingly (Lines 188–189).

Revision Plan

Reviewer #2, Minor point (4)

page 11, line 334

“kidney proximal tubule brush border cells (Cohen et al.) “The citation is missing the year of publication.

Response:

We have added the year of publication (Line 334).

Reviewer #2, Minor point (5)

Figure 3B

The Δ IDR1 panel is missing a small square indicating the region corresponding to the inset.

Response:

As pointed out by the Reviewer #3 in his/her minor comment (4), the number of mGFP-FTH1 puncta was significantly reduced in the Δ IDR1 mutant (Figures 3B, 3C). Thus, it would be more appropriate to remove the inset itself, rather than adding a small square corresponding to the inset. We have deleted the inset in the Δ IDR1 panel (Figure 3B).

Reviewer #2, Minor point (6)

Abstract

“both require NCOA4, which is thought to be an autophagy receptor for ferritin. However, the exact function of NCOA4 remains elusive.” I suggest rephrasing “the exact function of NCOA4 remains elusive” to something like “it is unclear whether NCOA4 only acts as an autophagy receptor in ferritin degradation”.

Response:

We have changed the text accordingly (Lines 21–23).

Reviewer #3, Major comment (3)

Fig. 2C: While the phenotype displayed looks clear, the figure would benefit from a nuclei counterstain, in order to assess whether a comparable number of cells is present in each field.

Response:

We have added the contours of cells based on the bright-field images that we have already taken.

C

FIP200 KO expressing mGFP-NCOA4

Revision Plan

Reviewer #3, Major comment (4)

Fig. 2D: Please add a loading control for the western blot, in order to assess whether the modulations of the proteins shown are relevant.

Response:

We have added the HSP90 blot as a loading control.

D

Reviewer #3, Major comment (5)

Fig. 3F: The FTH1 western blot shows three bands: the mGFP-FTH1 fusion band, the 'endogenous' FTH1 (lower arrowhead), and an even lower band. Can you clarify what the lowest band represents? Of note, it is not clear what band has been quantified by densitometry among the three different ones: please clarify. In relation to the modulation of FTH1 expression, a less exposed western blot would also help appreciating its differences by eye.

Response:

The lowest band is thought to be a degradation product of FTH1, which is also observed in WT HeLa cells without exogenous expression of mGFP-FTH1. Thus, only the endogenous FTH1 bands indicated by arrowheads in Figure 3F were used for quantification in Figure 3G. The statements, "Asterisk represents a putative degradation product of FTH1" and "The ratio of the endogenous FTH1 band intensities" were added to the figure legends to Figures 3F and 3G. In addition, the FTH1 blot in Figure 3F has been replaced with a less exposed one, as suggested.

F

Revision Plan

Reviewer #3, Major comment (7)

Fig 3H: For this western blot as well, less exposure time would make the results stand out more clearly.

Response:

The FTH1 blot in Figure 3H has been replaced with a less exposed one, as suggested.

Reviewer #3, Minor comment (1)

Fig. 1:

Please, clarify in the methods section how the mGFP-FTH1 cells were established. Is the expression of the transgene(s) constitutive or inducible? Similarly, please state the cell type used for each experiment in the relative figure legends.

Response:

We are sorry for the unclear description of experimental methods. All the mammalian cell lines were HeLa cells. mRuby3-RAB5^{Q79L} was doxycycline-induced and all the other exogenous proteins were stably expressed. We have added the information into the method section and corresponding figure legends.

Reviewer #3, Minor comment (2)

Fig. 2:

I would soften the conclusions relative to the experiments in Fig. 2C-D. In lines 128-134, the authors state 'Although this result suggests that FTH1, but not FTL, is required for the formation of NCOA4 condensate, knockdown of FTH1 also reduced the expression levels of mGFP-NCOA4, which made it difficult to evaluate the specific role of FTH1 (Figures 2C, 2D)'. As for the authors' admission, knock-down of FTH1 reduces the expression levels of mGFP-NCOA4, hence preventing from concluding much in regard to the direct involvement of FTH1 in the condensate formation. For this reason, we suggest to stress that the mGFP-NCOA4 reporter is affected by the knockdown of FTH1, with either oligo interference, and for this reason no clear conclusions can be drawn from this experiment. Alternatively, the authors can either strengthen this experiment (e.g. by means of different approaches or testing different oligo interference against FTH1) or remove it.

Revision Plan

Response:

We appreciate the reasonable indication. We have softened the conclusions and changed the sentences to correctly describe the conclusion drawn from this experiment to "Knockdown of FTL did not impede mGFP-NCOA4 puncta formation in FIP200 KO cells (Figures 2C, 2D), suggesting that FTL is not required for the formation of NCOA4 condensate. In contrast, knockdown of FTH1 inhibited mGFP-NCOA4 puncta formation. However, as it also reduced the expression levels of mGFP-NCOA4, it was difficult to evaluate the specific role of FTH1 (Figures 2C, 2D)" (Lines 129–132).

Reviewer #3, Minor comment (4)

Fig. 3C:

According to the quantification of Fig. 3B, the effect of introducing the delta IDR1 mutant is equally significant as delta N and delta IDR2. What is the reasoning for stating that the delta IDR1 only results in a partial rescue, while delta N and delta IDR2 cause a full rescue?

Response:

This comment is valid. As pointed out, puncta formation was significantly suppressed in the Δ IDR1 mutant as well as in the Δ N and Δ IDR2 mutants. In the previous version, we showed a relatively atypical cell in Figure 3B. We have revised the text (Lines 154–155) and replaced the Δ IDR1 panel in Figure 3B with a more typical one.

4. Description of analyses that authors prefer not to carry out

Reviewer #1, General comment

There are no major issues regarding the presented experiments. To better clarify the dual role of TAX1BP1, authors may consider to investigate the role of p62, NBR1, OPTN in the Ferritin-NCOA4 complex. If any of them is involved in lysosomal or endosomal sorting this may help to clarify the molecular mechanisms. p62 role in phase separation has been described and more in general these autophagy receptors have several common functions.

Response:

We are currently conducting a comprehensive study of how TAX1BP1 recruits ferritin-NCOA4 condensates to macroautophagy and endosomal microautophagy and whether p62, NBR1, OPTN, and NDP52 are involved in these pathways. As this topic is rather general and not specific to ferritin-NCOA4 condensates, we think that it would be beyond the scope of this short Report article focusing on ferritin-NCOA4 condensate formation. We would like to prepare it as another full story.

Reviewer #3, Major comment (8)

Fig. 5: The evidence that ferritin-NCOA4 condensates are engulfed into endosomes in a TAX1BP1-dependent manner is very interesting. Does this ability of TAX1BP1 rely on its noncanonical LIR-motif?

Response:

Aforementioned, we are currently conducting a more comprehensive study of how TAX1BP1 recruits ferritin-NCOA4 condensates to macroautophagy and endosomal microautophagy. It includes the role of ATG8 homologs and the noncanonical LIR-motif of TAX1BP1. We think that it would be beyond the scope of this short Report article focusing on ferritin-NCOA4 condensate formation.

Reviewer #3, Major comment (9) In addition, does ferritin engulfment into endosomes rely on the ESCRT-III system, as observed for other substrates of the endosomal microautophagy (<https://doi.org/10.1083/jcb.201711002>)?

Response:

This is an important point and we are also interested in it. However, we would like to focus on the behavior and physical properties of ferritin-NCOA4 condensates in this paper.

April 8, 2022

Re: JCB manuscript #202203102T

Prof. Noboru Mizushima
The University of Tokyo, Graduate School and Faculty of Medicine
Department of Biochemistry and Molecular Biology,
7-3-1 Hongo, Bunkyo-ku,
Tokyo 113-0033
Japan

Dear Prof. Mizushima,

Thank you for submitting your manuscript entitled "NCOA4 drives ferritin phase separation to facilitate macroferritinophagy and endosomal macroferritinophagy." We invite you to submit a revision as outlined in your revision plan.

GENERAL GUIDELINES:

Text limits: Character count for a Report is < 20,000, not including spaces. Count includes title page, abstract, introduction, the joint Results & Discussion, and acknowledgments. Count does not include materials and methods, figure legends, references, tables, or supplemental legends. Please note that Reports have a combined Results & Discussion section.

Figures: Reports may have up to 5 main text figures. To avoid delays in production, figures must be prepared according to the policies outlined in our Instructions to Authors, under Data Presentation, <https://jcb.rupress.org/site/misc/ifora.xhtml>. All figures in accepted manuscripts will be screened prior to publication.

IMPORTANT: It is JCB policy that if requested, original data images must be made available. Failure to provide original images upon request will result in unavoidable delays in publication. Please ensure that you have access to all original microscopy and blot data images before submitting your revision.

Supplemental information: There are strict limits on the allowable amount of supplemental data. Reports may have up to 3 supplemental figures. Up to 10 supplemental videos or flash animations are allowed. A summary of all supplemental material should appear at the end of the Materials and methods section.

Please note that JCB now requires authors to submit Source Data used to generate figures containing gels and Western blots with all revised manuscripts. This Source Data consists of fully uncropped and unprocessed images for each gel/blot displayed in the main and supplemental figures. Since your paper includes cropped gel and/or blot images, please be sure to provide one Source Data file for each figure that contains gels and/or blots along with your revised manuscript files. File names for Source Data figures should be alphanumeric without any spaces or special characters (i.e., SourceDataF#, where F# refers to the associated main figure number or SourceDataFS# for those associated with Supplementary figures). The lanes of the gels/blots should be labeled as they are in the associated figure, the place where cropping was applied should be marked (with a box), and molecular weight/size standards should be labeled wherever possible.

The typical timeframe for revisions is three to four months. While most universities and institutes have reopened labs and allowed researchers to begin working at nearly pre-pandemic levels, we at JCB realize that the lingering effects of the COVID-19 pandemic may still be impacting some aspects of your work, including the acquisition of equipment and reagents. Therefore, if you anticipate any difficulties in meeting this aforementioned revision time limit, please contact us and we can work with you to find an appropriate time frame for resubmission. Please note that papers are generally considered through only one revision cycle, so any revised manuscript will likely be either accepted or rejected.

Thank you for this interesting contribution to Journal of Cell Biology. You can contact us at the journal office with any questions, cellbio@rockefeller.edu or call (212) 327-8588.

Sincerely,

Hong Zhang, PhD
Monitoring Editor
Journal of Cell Biology

Dan Simon, PhD
Scientific Editor
Journal of Cell Biology

Full Revision

Manuscript number: RC-2022-01273

Corresponding author(s): Noboru Mizushima

1. General Statements [optional]

We would like to thank all three reviewers for their positive and constructive comments on our manuscript.

While Reviewer #1 has no major issues, he/she suggests further investigation on the role of p62, NBR1, and OPTN on ferritin turnover to better clarify the dual role of TAX1BP1. Also, Reviewer #3 asked about the mechanism of TAX1BP1 recruitment and the involvement of ESCRTs in his/her major comments #8 and #9. We are now conducting a comprehensive study to reveal the role of different types of autophagy adaptor proteins (including TAX1BP1) in microautophagy. As this topic is rather general and not specific to ferritin-NCOA4 condensates, we think that it would be beyond the scope of this short Report article focusing on ferritin-NCOA4 condensate formation. We would like to prepare it as another full story. In fact, during Referee cross-commenting, Reviewer #2 said, "It will probably take too long to perform all the additional experiments suggested by the three reviewers. It will be OK with me if the authors decide to only pursue some but not all suggested experiments" and Reviewer #3 said, "I agree, the paper quality is altogether high. In the case of my review, I consider the part regarding Figure 1 (expts. on endogenous targets) of the highest priority". We have addressed all the comments either experimentally or texturally.

After we posted this manuscript on bioRxiv on January 31st, 2022 and submitted it to Review Commons, Kazuhiro Iwai's group published a paper showing that NCOA4 forms fluid-like condensates that can include ferritin under prolonged iron-replete conditions (<https://doi.org/10.15252/embr.202154278>). We have cited and discussed this work in the Conclusion (Lines 291–296).

Full Revision

Responses to Reviewer #1

In the manuscript the authors proposed that ferritin particles undergo phase separation. Ferritin condensate with NCOA4 and can follow two different routes: lysosome-macroautophagy or endosomes-microautophagy. Ferritin-NCOA4-TAX1BP1 and LC3B binding determines the delivery to the lysosomes while the sorting to endosomes seems to be LC3B independent. The paper is well written and easy to follow. Experiments are properly controlled and the biochemistry, to explain the complexes formation and molecular interaction, is solid. There are no major issues regarding the presented experiments. To better clarify the dual role of TAX1BP1, authors may consider to investigate the role of p62, NBR1, OPTN in the Ferritin-NCOA4 complex. If any of them is involved in lysosomal or endosomal sorting this may help to clarify the molecular mechanisms. p62 role in phase separation has been described and more in general these autophagy receptors have several common functions.

Response:

We would like to thank Reviewer #1 for his/her positive and constructive comments on our manuscript. We understand that Reviewer #1 has no major issues in the current manuscript. Regarding Reviewer #1's additional issues for better understanding of the dual role of TAX1BP1, we are currently conducting a more comprehensive study of how TAX1BP1 recruits ferritin-NCOA4 condensates to macroautophagy and endosomal microautophagy and whether p62, NBR1, OPTN, and NDP52 are involved in these pathways. As this topic is rather general and not specific to ferritin-NCOA4 condensates, we think that it would be beyond the scope of this short Report article focusing on ferritin-NCOA4 condensate formation. We would like to prepare it as another full story.

Minor comment: check the figure format because some of them have been stretched likely during pdf assembly.

Response:

We thank Reviewer #1 for the suggestion. We have checked the figure format and revised the figures.

The manuscript provide a relative simple information of great value because it feels a gap in the Ferritin-NCOA4 (Ferritinophagy) field. For sure this finding is of potential interest for the autophagy community and also for researcher devoted to the study of human diseases linked to iron imbalance.

****Referees cross-commenting****

Full Revision

The paper is interesting and the experiments well performed. Addressing the major points should be feasible for the authors.

Response:

We thank Reviewer #1 for the generous and valuable comments.

Full Revision

Responses to Reviewer #2

In this study, the authors discover that NCOA4 mediates phase separation of ferritin particles. This function of NCOA4 requires the self interaction of NCOA4 through its N-terminal coiled coil region. Disrupting the self interaction of NCOA4 prevents the phase separation of ferritin particles and blocks ferritin degradation. The findings of this study shed important new light on the functions of NCOA4 in selective autophagy and reveal interesting connections between phase separation and selective autophagy.

Response:

We would like to thank Reviewer #2 for his/her positive comments on our manuscript.

Major point

In Figure 4F, the authors show that in NCOA4 KO cells, mGFP-FTH1 is no longer incorporated into endosomes. The authors draw the conclusion that “the formation of NCOA4-mediated condensates is required for the incorporation of ferritin into endosomes.” I think this conclusion can be strengthened by testing whether mutant forms of NCOA4 can rescue this phenotype.

Response:

As suggested by Reviewer #2, we examined whether mutant forms of NCOA4 rescued ferritin incorporation into endosomes in *NCOA4* KO cells. In accordance with our previous finding that the ΔN , $\Delta IDR2$, I56E, and L63R mutant forms could not rescue ferritin condensate formation (Figures 3B and 3C), all these mutants were not able to restore endosomal uptake of ferritin (Figure S3A). These results further strengthen our conclusion that the ferritin–NCOA4 condensate formation is required for endosomal microautophagy of ferritin. We have explained these data on Page 8, Lines 238–240.

Figure S3A. The ΔN , $\Delta IDR2$, I56E, and L63R mutants of NCOA4 are defective in ferritin incorporation into endosomes

NCOA4 KO HeLa cells expressing mGFP-FTH1 and FLAG-NCOA4 (WT or mutants) were treated with 2 $\mu\text{g}/\text{mL}$ doxycycline for 30 h to induce mRuby3-RAB5^{Q79L} expression and then observed by fluorescence microscopy.

Minor point (1)

Supplementary Figure S2: The predicted structure shown in the figure is a monomeric structure, perhaps because the software used for the prediction cannot predict homo-oligomeric structures. The AlphaFold-Multimer software can predict homo-oligomeric structures and may be worth trying.

Response:

We thank Reviewer #2 for this helpful suggestion. We used the AlphaFold-Multimer software and successfully predicted the homodimer of the N-terminal domain of NCOA4 as shown below. We have revised the text (Lines 173–174) and Figure S2B.

Figure S2B.

The structure of the N-terminal domain (residues 1–182) of NCOA4 (green and gray) is predicted by AlphaFold-Multimer. The putative self-interaction sites Ile-56 (magenta) and Leu-63 (red) are shown.

Minor point (2)

Figure 5G: In the model, the two subunits of an NCOA4 homodimer interact with two ferritin particles rather than interact with FTH1 molecules within a single ferritin particle. The authors may want to describe this particular aspect of the model and discuss the possible underlying mechanism.

Response:

We thank Reviewer #2 for clarifying this point. We think that an NCOA4 homodimer can interact with FTH1 either in the same particle or in different particles (bridging two particles). We drew only the latter (interparticle) in the scheme for simplicity, but it might be confusing. We have added the former type of interaction (intraparticle) in the model as well (Figure 5G).

Minor point (3)

Page 7, line 189: “these results suggest that condensate formation rather than NCOA4-ferritin binding itself is important for ferritin degradation.” I think a better way to summarize the results may be something like “these results suggest that NCOA4 dimerization-mediated condensate formation is important for ferritin degradation.”

Response:

Full Revision

We thank Reviewer #2 for this suggestion. We have changed the sentence accordingly (Lines 194–195).

Minor point (4)

Page 12, line 335: “kidney proximal tubule brush border cells (Cohen et al.)” The citation is missing the year of publication.

Response:

We have added the year of publication (Line 332).

Minor point (5)

Figure 3B: The Δ IDR1 panel is missing a small square indicating the region corresponding to the inset.

Response:

Regarding the data of the Δ IDR1 mutant, as Reviewer #3 pointed out in his/her minor comment (4), the number of mGFP-FTH1 puncta was significantly reduced in the Δ IDR1 mutant (Figures 3B, 3C). This is what we should emphasize. Therefore, it would be more appropriate to remove the inset, rather than add a small square corresponding to the inset. We have deleted the inset in the Δ IDR1 panel (Figure 3B).

Minor point (6)

Abstract: “both require NCOA4, which is thought to be an autophagy receptor for ferritin. However, the exact function of NCOA4 remains elusive.” I suggest rephrasing “the exact function of NCOA4 remains elusive” to something like “it is unclear whether NCOA4 only acts as an autophagy receptor in ferritin degradation”.

Response:

We have changed the sentence according to this reviewer’s suggestion (Lines 24–26).

This study uncovers a previous unknown role of NCOA4 in selective autophagy. The findings reported in this manuscript broaden the understandings of the mechanisms and significances of phase separation in selective autophagy. This paper should be of interest to scientists studying autophagy and phase separation. My expertise is in selective autophagy.

Full Revision

****Referees cross-commenting****

I think the points raised by the other two reviewers are all valid and worthy of consideration by the authors. It will probably take too long to perform all the additional experiments suggested by the three reviewers. It will be OK with me if the authors decide to only pursue some but not all suggested experiments.

Response:

We thank Reviewer #2 for the generous and valuable comments.

Full Revision

Responses to Reviewer #3

The manuscript from Ohshima and colleagues describes liquid-liquid phase separation (LLPS) of ferritin, and its impact on macro- and endosomal micro-autophagy of ferritin particles (namely macro-ferritinophagy and micro-ferritinophagy, respectively). Of note, in this manuscript, endosomal micro-autophagy of ferritin is described for the first time. From a mechanistic viewpoint, the ferritin-cargo receptor NCOA4 drives ferritin LLPS through homo-dimerisation and interaction with FTH1, which is a constituent of ferritin particles. Interestingly, this phenomenon is functional to ferritin degradation through both macroferritinophagy and microferritinophagy. Also, the degradation of ferritin-NCOA4 condensates occurs in a TAX1BP1-dependent manner, in both macroferritinophagy and microferritinophagy. Altogether, the study is technically elegant and sound. It relies on exogenous reporters monitored through imaging/live-imaging, several Knock-Out (KO) models, and an elegant NCOA4-FTH1 interaction assay carried out in yeast, which does not express yeast homologues for the proteins of interest. Also, the reported results are of great interest for the field of selective autophagy. However, small implementations are suggested to further strengthen the results. In particular, the data reporting on ferritin being a substrate of endosomal microautophagy should be better related to the current knowledge of the pathway.

Response:

We would like to thank Reviewer #3 for his/her positive and valuable suggestions on our manuscript.

Major comment (1)

Fig. 1: As many other figures along the manuscript, figure 1 relies on the imaging of exogenous reporters: mGFP-FTH1/FTL. While these reporters look reliable on the basis of their localization and changes upon autophagy modulation, a co-staining of mGFP and an endogenous marker of ferritin is suggested, in order to verify the proper localization of the exogenous reporters (i.e. mGFP-FTH1 cells stained with an antibody against endogenous FTL, and mGFP-FTL cells stained with an antibody against endogenous FTH1).

Response:

We also appreciate the importance of the proper localization of endogenous ferritin. We confirmed by immunostaining using an anti-FTL monoclonal antibody that endogenous FTL colocalized well with mGFP-FTH1 and Halo-NCOA4 puncta (Figures S1A, S1D). These results validate the use of the exogenous FTH1 reporter. We have tried several anti-FTH1 antibodies, but none of them worked in immunostaining. We have modified the text accordingly (Lines 95–96 and 123-124).

Figure S1A and S1D. Exogenously expressed mGFP-FTH1 and Halo-NCOA4 co-localized with endogenous FTL

WT HeLa cells expressing mGFP-FTH1 (A) or Halo-NCOA4 (D) were grown in DMEM, treated with 100 nM HaloTag SaraFluor 650T ligand (D), and then subjected to immunostaining with anti-FTL monoclonal antibody.

Major comment (2)

Fig. 1E-G: This type of experiments is crucial in order to determine the LLPS properties of ferritin. Therefore, I would suggest to perform it also in mGFP-FTL expressing cells, to further strengthen the data already shown for mGFP-FTH1.

Response:

We had already tried the FRAP experiment using mGFP-FTL, but mGFP-FTL puncta were smaller and moved faster than mGFP-FTH1 puncta. This is probably because FTH1 is more important to generate large condensates. For this reason, it is difficult to track and quantify the mGFP-FTL signal after photobleaching. In the revised manuscript, we show only qualitative time-lapse images of mGFP-FTL puncta exhibiting fluorescence recovery after photobleaching (Figure S1C). We have modified the text accordingly (Lines 110–111)

Figure S1C. mGFP-FTL puncta showed fluorescence recovery after photobleaching

FRAP analyses of the ferritin condensates. WT HeLa cells expressing mGFP-FTL were treated with 100 $\mu\text{g/mL}$ FAC for 24 h followed by 50 μM DFO for 2 h, and subjected to FRAP analyses (4-s intervals for 128 s). Three mGFP-FTL puncta are shown. Scale bars, 2 μm .

Major comment (3)

Fig. 2C: While the phenotype displayed looks clear, the figure would benefit from a nuclei counterstain, in order to assess whether a comparable number of cells is present in each field.

Response:

We have added the contours of cells based on the bright-field images that we have already taken (Figures 2C).

Major comment (4)

Fig. 2D: Please add a loading control for the western blot, in order to assess whether the modulations of the proteins shown are relevant.

Response:

Full Revision

We have added the blot of the loading control HSP90, which we have already taken (new Fig. 2E).

Major comment (5)

Fig. 3F: The FTH1 western blot shows three bands: the mGFP-FTH1 fusion band, the ‘endogenous’ FTH1 (lower arrowhead), and an even lower band. Can you clarify what the lowest band represents? Of note, it is not clear what band has been quantified by densitometry among the three different ones: please clarify. In relation to the modulation of FTH1 expression, a less exposed western blot would also help appreciating its differences by eye.

Response:

The lowest band is thought to be a degradation product of FTH1, which is also observed in WT HeLa cells without exogenous expression of mGFP-FTH1. Only the endogenous FTH1 bands, which are indicated by the arrowhead in Figure 3F, were used for quantification in Figure 3G. To clarify these points, we have added the statements “asterisk represents a putative degradation product of FTH1” and “the endogenous FTH1 band intensities” to the figure legends to Figures 3F and 3G, respectively. In addition, the FTH1 blot in Figure 3F has been replaced with a less exposed one, as suggested, and quantification in Figure 3G has also been replaced using this blot.

Full Revision

F

G

Major comment (6)

Since the whole ferritin is supposed to be substrate of macroferritinophagy and microferritinophagy, it would be more solid to rely on both ferritin markers rather than on only one. For this reason, it should be investigated how FTL is modulated upon FAC and DFO treatments, by means western blot analysis and the relative densitometric quantification.

Response:

We repeated the western blot analysis of FTL and added the results in the revised manuscript (Figures 3F [see above] and 3H). However, the changes in the expression level of FTL were not accordant with that of FTH1, consistent with the previous reports which demonstrated that FTH1 and FTL are differentially regulated by iron on a transcriptional level (Cairo et al., *Biochem. Biophys. Res. Commun.*, 1985, White & Munro, *J. Biol. Chem.*, 1988, Muckenthaler et al., *Blood*, 2003). Considering that only FTH1 has ferroxidase activity, that FTH1 is the dominant subunit in HeLa cells, and that most of the published reports investigate FTH1 levels to assess the effects of iron on ferritin, we decided to display panels of FTH1 and FTL, while using FTH1 bands for quantification.

Full Revision

Major comment (7)

Fig 3H: For this western blot as well, less exposure time would make the results stand out more clearly.

Response:

The FTH1 blot in Figure 3H (see above) has been replaced with a less exposed one, and quantification in Figure 3I has also been replaced using this blot.

Major comment (8)

Fig. 5: The evidence that ferritin-NCOA4 condensates are engulfed into endosomes in a TAX1BP1-dependent manner is very interesting. Does this ability of TAX1BP1 rely on its noncanonical LIR-motif?

Response:

We thank Reviewer #3 for this constructive suggestion. As aforementioned, we are currently conducting a more comprehensive study of how TAX1BP1 recruits ferritin-NCOA4 condensates to macroautophagy and endosomal microautophagy. It includes the role of ATG8 homologs and the noncanonical LIR-motif of TAX1BP1 and other autophagy receptors. We think that it would be beyond the scope of this short Report article focusing on ferritin-NCOA4 condensate formation.

Full Revision

Major comment (9)

In addition, does ferritin engulfment into endosomes rely on the ESCRT-III system, as observed for other substrates of the endosomal microautophagy (<https://doi.org/10.1083/jcb.201711002>)?

Response:

We think that it is likely dependent on the ESCRT machinery. However, we would like to focus on the behavior and physical properties of ferritin-NCOA4 condensates in this short paper.

Major comment (10)

Also, what is the hypothesis regarding the endpoint of the endosomal route? Are the endosomes delivered to the lysosome, as one would expect if this is endosomal microautophagy. This should be properly tested by co-localisation of ferritin-NCOA4 containing endosomes and lysosomes, for example by using an endogenous marker of the endosomes (e.g. CD63, as previously reported here <https://doi.org/10.1083/jcb.201711002>) and the lysosomal membrane marker LAMP1, in order to strengthen the evidence of ferritin-NCOA4 condensates as substrates for endosomal microautophagy.

Response:

To track the delivery of ferritin-NCOA4 condensates to lysosomes, we used a pulse-labelable Halo-NCOA4 reporter because HaloTag becomes stable in lysosomes after ligand binding (Rudinskiy et al. Mol Biol Cell 2022, Yim et al. bioRxiv 2022). In WT HeLa cells, fluorescent signals of Halo-NCOA4 labeled with the SaraFluor 650T ligand for 24 h well colocalized with LysoTracker Red (Figures S3B, S3C). This is considered to be the result of both macro- and micro-autophagy. To specifically track the microautophagy pathway, we used macroautophagy-deficient *FIP200* KO cells and found that colocalization between Halo-NCOA4 and LysoTracker was reduced compared with WT cells. However, there remained some colocalization signals, which should represent the result of ferritin microautophagy. In *TAX1BP1* KO cells, in which both pathways are blocked, Halo-NCOA4 did not colocalize with LysoTracker signals at all. From these results, we assume that the lysosome is at least one of the endpoints of endosomal microautophagy of ferritin-NCOA4 condensates. We have modified the text accordingly (Lines 269–280). However, as we mention in the Conclusion, it is also possible that ferritin-NCOA4 condensates are secreted via endosomes (Lines 312–317).

Figure S3B and S3C. TAX1BP1 is required for both macroautophagy and microautophagy of ferritin–NCOA4 condensates

(B) WT, *FIP200* KO, and *TAX1BP1* KO HeLa cells expressing Halo-NCOA4 were treated with 20 nM HaloTag SaraFluor 650T ligand for 24 h in DMEM. The cells were further treated with LysoTracker Red for 15 min in DMEM and then observed by fluorescence microscopy. (C) Fluorescence intensity of Halo-NCOA4 per LysoTracker Red-positive structure in (B) was quantified (n = 2235–3626 LysoTracker Red-positive structures). Differences were statistically analyzed by Welch's t-test with Holm-Sidak method for multiple comparison.

Major comment (11)

In addition, it would be interesting to assess the contribution of endosomal microautophagy to ferritin degradation both in basal condition and upon induction (i.e. upon FAC followed by DFO treatment, similarly to what was done in fig. 3F). This can be tested by disruption of the multivesicular body dynamics via U18666A treatment, as previously shown (<https://doi.org/10.1083/jcb.201711002>).

Response:

This is an interesting experiment. Using *FIP200* KO cells in which we could specifically track microautophagy of ferritin (Figures S3B, S3C), we assessed the effect of the cholesterol transport inhibitor U18666A. However, the accumulation of Halo-NCOA4 signals in lysosomes was still observed in U18666A-treated *FIP200* KO cells. Moreover, U18666A affected lysosomal morphology, making it difficult to evaluate fluorescent signals properly. We need to refine experimental conditions or establish other methods to evaluate the contribution of ferritin macro- and micro-autophagy properly, which we would like to pursue in the future.

Full Revision

Major comment (12)

Along the same line, the authors could investigate whether microferritinophagy still occur upon upstream inhibition of autophagy (e.g. by means of ATG5/7 KO or FIP200 KO, or by treating cells with an ULK1 inhibitor)? In fact, if endosomal microautophagy contributes to ferritin degradation, partial degradation should still be observed under these conditions. Also, microferritinophagy-mediated degradation should be fully abolished upon MVB disruption and/or upon Vps34 inhibition, combined to upstream autophagy inhibition (<https://doi.org/10.1083/jcb.201711002>).

Response:

As we mentioned in our response to Major comment (10), ferritin microautophagy likely occurs in macroautophagy-deficient *FIP200* KO cells (Figures S3B, S3C).

Minor comment (1)

Fig. 1: Please, clarify in the methods section how the mGFP-FTH1 cells were established. Is the expression of the transgene(s) constitutive or inducible? Similarly, please state the cell type used for each experiment in the relative figure legends.

Response:

We are sorry for the unclear description of experimental methods. All the mammalian cell lines used in this study were generated using HeLa cells. mRuby3-RAB5^{Q79L} was expressed in a doxycycline-induced manner, and all the other exogenous proteins were expressed stably. We have added that information into the method section (Lines 425–427) and figure legends.

Minor comment (2)

Fig. 2: I would soften the conclusions relative to the experiments in Fig. 2C-D. In lines 130-133, the authors state 'Although this result suggests that FTH1, but not FTL, is required for the formation of NCOA4 condensate, knockdown of FTH1 also reduced the expression levels of mGFP-NCOA4, which made it difficult to evaluate the specific role of FTH1 (Figures 2C, 2D (2D, 2E in the revised version))'. As for the authors' admission, knock-down of FTH1 reduces the expression levels of mGFP-NCOA4, hence preventing from concluding much in regard to the direct involvement of FTH1 in the condensate formation. For this reason, we suggest to stress that the mGFP-NCOA4 reporter is affected by the knockdown of FTH1, with either oligo interference, and for this reason no clear conclusions can be drawn from this experiment. Alternatively, the authors can either strengthen this experiment (e.g. by means of different approaches or testing different oligo interference against FTH1) or remove it.

Response:

We appreciate the reasonable indication. We have softened the conclusions and changed the sentences to correctly describe the conclusion drawn from this experiment

Full Revision

to “Knockdown of FTL did not impede mGFP-NCOA4 puncta formation in *FIP200* KO cells (Figures 2D, 2E), suggesting that FTL is not required for the formation of NCOA4 condensate. In contrast, knockdown of FTH1 inhibited mGFP-NCOA4 puncta formation. However, as it also reduced the expression levels of mGFP-NCOA4, it was difficult to evaluate the specific role of FTH1 (Figures 2D, 2E)” (Lines 135–138).

Minor comment (3)

Quantification of the imaging experiments in Fig. 2 should be performed, i.e. number of ferritin-NCOA4 condensates in WT vs. KO cells. Also, the number of cells analyzed should be reported in the figure legend.

Response:

We have counted the number of ferritin-NCOA4 condensates in WT and KO cells and presented the results in Figure 2C. The number of cells analyzed has been added to the legend of Figure 2C.

Minor comment (4)

Fig. 3C: According to the quantification of Fig. 3B, the effect of introducing the delta IDR1 mutant is equally significant as delta N and delta IDR2. What is the reasoning for stating that the delta IDR1 only results in a partial rescue, while delta N and delta IDR2 cause a full rescue?

Response:

This comment is valid. As pointed out, puncta formation was significantly suppressed in the Δ IDR1 mutant as well as in the Δ N and Δ IDR2 mutants. In the previous version, we showed a relatively atypical cell in Figure 3B. We have revised the text (Lines 160–161) and replaced the Δ IDR1 panel in Figure 3B with a more typical one.

Recently, liquid-liquid phase separation (LLPS) has been reported to impact cargo selection in selective autophagy mediated by macroautophagy. At present, nothing is known about the relationship between LLPS, ferritin degradation and endosomal microautophagy. This manuscript describes ferritin to have LLPS properties, and to be a substrate of endosomal microautophagy. Hence, this paper fills a gap in our knowledge and suggests that LLPS is relevant to endosomal microautophagy, as well as to ferritin

Full Revision

homeostasis. Of note, increased ferritin secretion is observed in several diseases. As a result, the present study is of potential interest for the clinically relevant features of such diseases.

****Referees cross-commenting****

I agree, the paper quality is altogether high. In the case of my review, I consider the part regarding Figure 1 (expts. on endogenous targets) of the highest priority.

Response:

We appreciate Reviewer #3 for constructive suggestions. We have added the data of the highest priority experiments in Figure 1.

References

- Cairo, G., F. Bernuzzi, and S. Recalcati. 2006. A precious metal: Iron, an essential nutrient for all cells. *Genes Nutr.* 1:25-39.
- Muckenthaler, M., A. Richter, N. Gunkel, D. Riedel, M. Polycarpou-Schwarz, S. Hentze, M. Falkenhahn, W. Stremmel, W. Ansorge, and M.W. Hentze. 2003. Relationships and distinctions in iron-regulatory networks responding to interrelated signals. *Blood.* 101:3690-3698.
- White, K., and H.N. Munro. 1988. Induction of ferritin subunit synthesis by iron is regulated at both the transcriptional and translational levels. *J Biol Chem.* 263:8938-8942.
- Rudinskiy, M., T.J. Bergmann, Molinari, M. 2022. Quantitative and time-resolved monitoring of organelle and protein delivery to the lysosome with a tandem fluorescent Halo-GFP reporter. *Mol Biol Cell* 33:ar57
- Yim, W.W., H. Yamamoto, and N. Mizushima. 2022. A pulse-chasable reporter processing assay for mammalian autophagic flux with HaloTag. *bioRxiv*. doi: <https://doi.org/10.1101/2022.04.13.488149>

July 19, 2022

RE: JCB Manuscript #202203102R

Prof. Noboru Mizushima
The University of Tokyo, Graduate School and Faculty of Medicine
Department of Biochemistry and Molecular Biology,
7-3-1 Hongo, Bunkyo-ku,
Tokyo 113-0033
Japan

Dear Prof. Mizushima,

Thank you for submitting your revised manuscript titled "NCOA4 drives ferritin phase separation to facilitate macroferritinophagy and microferritinophagy." We would be happy to publish your paper in JCB pending final revisions necessary to meet our formatting guidelines (see details below).

A. MANUSCRIPT ORGANIZATION AND FORMATTING:

- 1) Text limits: Character count for Reports is < 20,000, not including spaces. Count includes title page, abstract, introduction, results, discussion, and acknowledgments. Count does not include materials and methods, figure legends, references, tables, or supplemental legends.
- 2) Figures: Reports may have up to 5 main text figures. Scale bars must be present on all microscopy images, including inset magnifications. Molecular weight or nucleic acid size markers must be included on all gel electrophoresis. Please add a scale bar for the time-lapse images in Figure 4C.
- 3) Statistical analysis: Error bars on graphic representations of numerical data must be clearly described in the figure legend. The number of independent data points (n) represented in a graph must be indicated in the legend. Statistical methods should be explained in full in the materials and methods. For figures presenting pooled data the statistical measure should be defined in the figure legends. Please also be sure to indicate the statistical tests used in each of your experiments (both in the figure legend itself and in a separate methods section) as well as the parameters of the test (for example, if you ran a t-test, please indicate if it was one- or two-sided, etc.). Also, if you used parametric tests, please indicate if the data distribution was tested for normality (and if so, how). If not, you must state something to the effect that "Data distribution was assumed to be normal but this was not formally tested."
- 4) Materials and methods: Should be comprehensive and not simply reference a previous publication for details on how an experiment was performed. Please provide full descriptions (at least in brief) in the text for readers who may not have access to referenced manuscripts. The text should not refer to methods "...as previously described." Please indicate the type of membrane used for immunoblotting.
- 5) For all cell lines, vectors, constructs/cDNAs, etc. - all genetic material: please include database / vendor ID (e.g., Addgene, ATCC, etc.) or if unavailable, please briefly describe their basic genetic features, even if described in other published work or gifted to you by other investigators (and provide references where appropriate). Please be sure to provide the sequences for all of your oligos: primers, si/shRNA, RNAi, gRNAs, etc. in the materials and methods. You must also indicate in the methods the source, species, and catalog numbers/vendor identifiers (where appropriate) for all of your antibodies, including secondary. If antibodies are not commercial please add a reference citation if possible.
- 6) Microscope image acquisition: The following information must be provided about the acquisition and processing of images:
 - a. Make and model of microscope
 - b. Type, magnification, and numerical aperture of the objective lenses
 - c. Temperature
 - d. Imaging medium
 - e. Fluorochromes
 - f. Camera make and model
 - g. Acquisition software

h. Any software used for image processing subsequent to data acquisition. Please include details and types of operations involved (e.g., type of deconvolution, 3D reconstitutions, surface or volume rendering, gamma adjustments, etc.).

7) References: There is no limit to the number of references cited in a manuscript. References should be cited parenthetically in the text by author and year of publication. Abbreviate the names of journals according to PubMed.

8) Supplemental materials: There are strict limits on the allowable amount of supplemental data. Reports may have up to 3 supplemental figures and 10 videos. Please also note that tables, like figures, should be provided as individual, editable files. A summary of all supplemental material should appear at the end of the Materials and methods section. Please include one brief sentence per item.

9) Video legends: Should describe what is being shown, the cell type or tissue being viewed (including relevant cell treatments, concentration and duration, or transfection), the imaging method (e.g., time-lapse epifluorescence microscopy), what each color represents, how often frames were collected, the frames/second display rate, and the number of any figure that has related video stills or images.

10) eTOC summary: A ~40-50 word summary that describes the context and significance of the findings for a general readership should be included on the title page. The statement should be written in the present tense and refer to the work in the third person. It should begin with "First author name(s) et al..." to match our preferred style.

11) Conflict of interest statement: JCB requires inclusion of a statement in the acknowledgements regarding competing financial interests. If no competing financial interests exist, please include the following statement: "The authors declare no competing financial interests." If competing interests are declared, please follow your statement of these competing interests with the following statement: "The authors declare no further competing financial interests."

12) A separate author contribution section is required following the Acknowledgments in all research manuscripts. All authors should be mentioned and designated by their first and middle initials and full surnames. We encourage use of the CRediT nomenclature (<https://casrai.org/credit/>).

13) ORCID IDs: ORCID IDs are unique identifiers allowing researchers to create a record of their various scholarly contributions in a single place. At resubmission of your final files, please consider providing an ORCID ID for as many contributing authors as possible.

14) Please note that JCB now requires authors to submit Source Data used to generate figures containing gels and Western blots with all revised manuscripts. This Source Data consists of fully uncropped and unprocessed images for each gel/blot displayed in the main and supplemental figures. Since your paper includes cropped gel and/or blot images, please be sure to provide one Source Data file for each figure that contains gels and/or blots along with your revised manuscript files. File names for Source Data figures should be alphanumeric without any spaces or special characters (i.e., SourceDataF#, where F# refers to the associated main figure number or SourceDataFS# for those associated with Supplementary figures). The lanes of the gels/blots should be labeled as they are in the associated figure, the place where cropping was applied should be marked (with a box), and molecular weight/size standards should be labeled wherever possible. Source Data files will be directly linked to specific figures in the published article.

B. FINAL FILES:

****The license to publish form must be signed before your manuscript can be sent to production. A link to the electronic license to publish form will be sent to the corresponding author only. Please take a moment to check your funder requirements before choosing the appropriate license.****

Thank you for this exciting contribution, we look forward to publishing your paper in Journal of Cell Biology.

Sincerely,

Hong Zhang, PhD
Monitoring Editor
Journal of Cell Biology

Dan Simon, PhD
Scientific Editor
Journal of Cell Biology